# EMOTIONTHINKER: PROSODY-AWARE REINFORCEMENT LEARNING FOR EXPLAINABLE SPEECH EMOTION REASONING

**Dingdong Wang**[1,2]*, **Shujie Liu**[2], **Tianhua Zhang**[1], **Youjun Chen**[1], **Jinyu Li**[2], **Helen Meng**[1]
[1]The Chinese University of Hong Kong
[2]Microsoft Corporation
dingdongwang@link.cuhk.edu.hk

## ABSTRACT

Emotional information in speech plays a unique role in multimodal perception. However, current Speech Large Language Models (SpeechLLMs), similar to conventional speech emotion recognition (SER) systems, still treat emotion understanding as a simple classification problem. This provides limited interpretability of predictions, while leaving the LLMs' expressive and reasoning capabilities underutilized. In this work, we take the first step to reformulate SER as a deep reasoning problem through reinforcement learning (RL). We propose **Emotion-Thinker**, which is designed to generate accurate emotion predictions with interpretable explanations grounded in fine-grained acoustic cues. To achieve this, we first construct **EmotionCoT-35K**, an emotional reasoning dataset with Chain-of-Thought annotations and detailed captions. Second, we observe that current SpeechLLMs exhibit weak prosody perception, whereas prosodic cues constitute fundamental signals for interpreting emotions. To address this, we develop the prosody-enhanced foundation model EmotionThinker-Base, and demonstrate that prosody enhancement improves emotion understanding. Third, we introduce **G**roup-**R**elative-**P**olicy-**O**ptimization with **P**rogressive-**T**rust-aware-**R**easoning-Reward (**GRPO-PTR**) for RL. Different from standard GRPO, which relies only on rule-based outcome rewards, GRPO-PTR progressively introduces reasoning reward, dynamically adjusts it with a trustworthiness weight reflecting the alignment between reasoning and outcome, and evaluates the overall reasoning quality with a reward model based on multi-dimensional criteria. EmotionThinker outperforms previous state-of-the-art evaluation models both in emotion accuracy and explanation quality, advancing SER toward interpretable multimodal reasoning. Project page: https://github.com/dingdongwang/EmotionThinker

## 1 INTRODUCTION

Advances in speech large language models (SpeechLLMs) have demonstrated impressive performance across various speech downstream tasks (Arora et al., 2025; Su et al., 2025; Ji et al., 2024a). Among them, speech emotion recognition (SER) is particularly important for human–computer interaction and affective computing systems (Wani et al., 2021; El Ayadi et al., 2011). Nevertheless, current SpeechLLMs largely inherit the paradigm of conventional SER systems, treating emotion understanding as a categorical classification problem (e.g., assigning a speaker to a discrete emotion label). Such formulations limit interpretability, while leaving the expressive and reasoning capabilities of multimodal LLMs underutilized. This gives rise to one intuitive thought: *Can SpeechLLMs reason like humans about "why" they make emotional judgments?*

Recent efforts have explored enhancing emotion explainability through supervised fine-tuning (SFT) (Xu et al., 2024; Liang et al., 2024; Zhang & Poellabauer; Wang et al., 2025c; Thimonier et al., 2025; Chen et al., 2025b; Zhang et al., 2025), prompting models to generate predictions alongside acoustic captions. However, these methods primarily operate at the descriptive level, failing to bridge

---

*Work done during an internship at Microsoft.

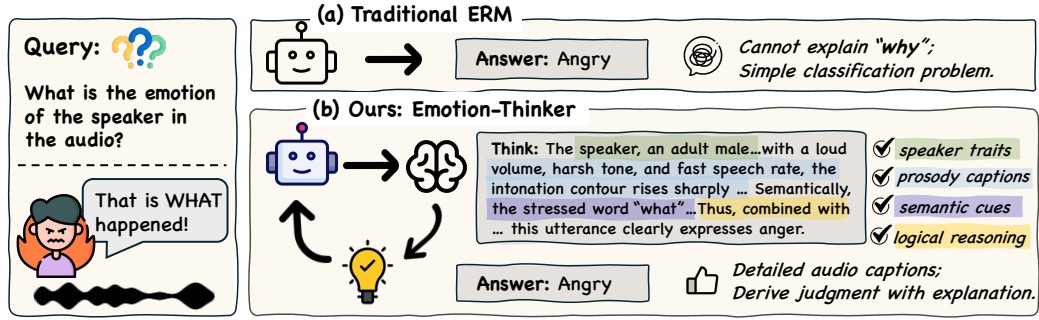

Figure 1: EmotionThinker generates emotion predictions with explanatory reasoning by leveraging speaker traits, prosody, semantics, and logic, in contrast to traditional models that provide only categorical outputs.

acoustic observations with explanatory reasoning about their emotional significance. In this work, we take the first step towards Reinforcement Learning (RL)-based explainable emotion reasoning. To this end, we identify three key challenges: (i) Scarcity of high-quality datasets: existing emotion corpora lack fine-grained acoustic annotations necessary for reasoning supervision; (ii) Weak prosody perception in foundation models: current SpeechLLMs struggle with acoustic cues—particularly prosody (e.g., stress, intonation, pitch), which plays a crucial role in conveying emotions; (iii) Limitations of standard rule-based RL rewards: optimizing only rule-based rewards (e.g., outcome accuracy) is insufficient for supervising reasoning quality during RL. As a result, the model may converge to sub-optimal reasoning strategies that yield the correct outcome but undermine trustworthiness.

To address these challenges, we propose **EmotionThinker**, an RL–enhanced SpeechLLM framework that offers the following advantages: (1) higher emotion recognition accuracy; (2) deep reasoning ability to integrate emotion-related cues for justification; (3) fine-gained audio caption covering speaker traits, prosodic cues and semantic information. Specifically, in the first stage, we curate **EmotionCoT-35K**, a high-quality Chain-of-Thought (CoT) dataset synthesized from open-source emotion datasets, which includes detailed acoustic cues for emotion reasoning. In this stage, we develop an automated annotation pipeline that extracts speaker traits (gender and age group), prosodic features (e.g., gender, age, pitch, speech rate, energy, stress, intonation), and transcription content. We use this pipeline to construct CoT training data via carefully designed prompts. In the second stage, prior to RL training, we construct EmotionThinker-Base, a prosody-enhanced foundation model based on Qwen2.5-Omni-7B (Xu et al., 2025b). We find that, as prosodic signals are core carriers of emotional intent, enhancing prosody perception allows the model to effectively ground its reasoning in affective cues and further improve emotion accuracy.

Finally, during RL training, to address the challenge of open-ended reasoning process, we propose **G**roup-**R**elative-**P**olicy-**O**ptimization with **P**rogressive-**T**rust-Aware **R**easoning-Reward (**GRPO-PTR**), a reward scheme that measures the intermediate reasoning quality across multiple dimensions. In this stage, we progressively introduce a thinking reward model trained on annotated reasoning responses of diverse quality. This reward model evaluates the reasoning process based on criteria such as factual alignment and interpretative quality. Furthermore, to mitigate the risk of reward hacking, we incorporate an additional trustworthiness weight that evaluates reasoning reward reliability across a group of samples for a given query. This weight penalizes situations when high reasoning rewards are mistakenly assigned to incorrect answers, helping to suppress spurious reward signals and ensure reasoning-outcome alignment. Experiments show that GRPO-PTR is effective for emotion reasoning, surpassing rule-based RL baselines both in reasoning quality and emotion accuracy. Building on this framework, EmotionThinker consistently outperforms 16 open-source SpeechLLMs across multiple benchmarks, delivering higher recognition and reasoning ability.

## 2 RELATED WORK

**Reasoning in SpeechLLMs.** Recent research on SpeechLLMs for emotion understanding can be broadly categorized into three directions: (1) General-purpose SpeechLLMs such as SALMONN (Tang et al., 2023), Kimi-Audio (KimiTeam et al., 2025), and Qwen-Audio series (Chu et al., 2023; 2024; Xu et al., 2025b) that treat emotion recognition as one of their downstream

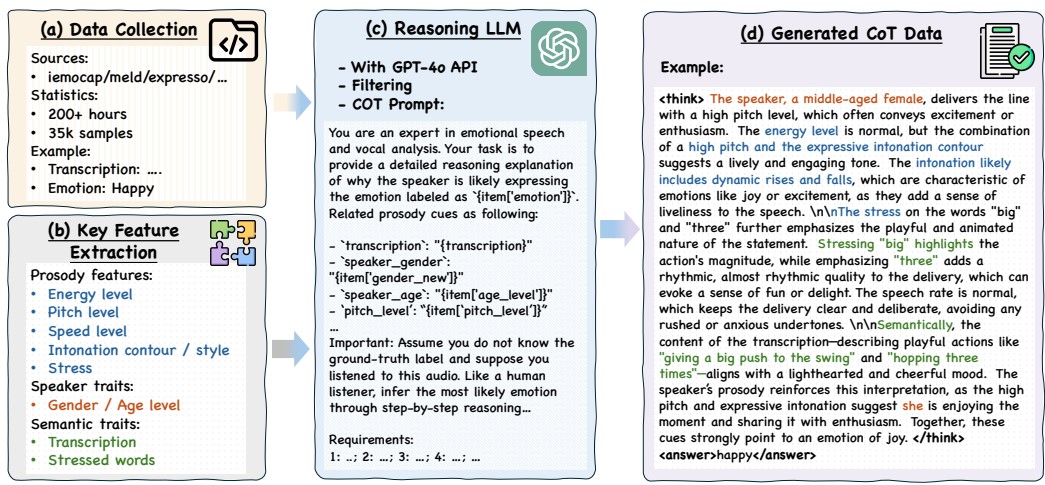

Figure 2: EmotionCoT-35K data curation pipeline.

evaluation tasks; (2) Empathetic dialogue systems (Xue et al., 2024; Chen et al., 2025a; Lin et al., 2024) (e.g., OSUM-Echat (Geng et al., 2025)) that utilize emotion recognition as a prerequisite for generating contextually appropriate responses with high emotional intelligence; (3) Descriptive emotion caption models (Xu et al., 2024; Liang et al., 2024; Zhang & Poellabauer; Wang et al., 2025c; Thimonier et al., 2025; Chen et al., 2025b; Zhang et al., 2025), which describe emotions in natural language alongside contextual information such as transcription and acoustic features. However, previous work shares a fundamental limitation: they focus primarily on improving emotion classification accuracy through architecture design, while lacking deep reasoning capabilities. Even caption-based approaches that attempt to enhance interpretability still suffer from limited granularity and lack causal links. They provide only surface-level descriptions without establishing systematic connections between acoustic features and emotional inferences. These gaps highlight the need for SpeechLLMs that move beyond categorical prediction to deliver deeper reasoning.

**Multimodal Large Reasoning Models.** Recent progress in reinforcement learning (RL) has highlighted its strong potential for enhancing reasoning capabilities across diverse domains (Shao et al., 2024; Guo et al., 2025; Wang et al., 2025a; Xu et al., 2025a). Notably, DeepSeek-R1 (Guo et al., 2025) demonstrated that RL with simple rule-based rewards can effectively incentivize robust reasoning without dense supervision, offering a scalable and efficient paradigm. Building on these advances, vision-language models (VLMs) have achieved remarkable success in multimodal reasoning (Wang et al., 2025d; Huang et al., 2025; Shen et al., 2025), such as in visual quality assessment (Wu et al., 2025) and image generation (Tong et al., 2025). However, the effectiveness of RL remains largely underexplored in the speech domain. In particular, speech emotion reasoning presents unique modality-specific challenges. Accurate emotion understanding requires integrating diverse acoustic cues, especially prosodic features such as stress, tone, and intonation. Yet outcome-based reward (Guo et al., 2025) alone cannot ensure these signals are correctly and comprehensively captured throughout the reasoning process. To achieve trustworthy emotion reasoning, additional mechanisms are needed to supervise intermediate reasoning steps and ensure their alignment with the final predictions.

## 3 METHOD

We propose a three-stage framework to equip EmotionThinker with explainable speech emotion recognition (SER) capabilities via explicit reasoning. First, we construct EmotionCoT-35K, a training dataset with fine-grained prosodic and reasoning annotations. Next, we develop EmotionThinker-Base based on Qwen2.5-Omni-7B (Xu et al., 2025b) by prosody-enhanced supervised fine-tuning and the cold-start training. Finally, we apply reinforcement learning with our proposed GRPO—PTR strategy to progressively refine reasoning quality while maintaining emotional consistency.

## 3.1 EMOTIONCOT-35K

**Overall information.** Existing speech emotion datasets are limited in offering reasoning-style supervision and detailed prosody-aware annotations. To facilitate explainable SER, we construct **EmotionCoT-35K**, a richly annotated training dataset of 35,000 speech–reasoning pairs spanning about 200 hours of audio. Each sample is annotated with an emotion label from nine common categories (Neutral, Happy, Sad, Angry, Contempt/Disgust, Confused, Whisper, Surprise, Fear). The dataset is sourced from IEMOCAP (Busso et al., 2008), MELD (Poria et al., 2019), Expresso (Nguyen et al., 2023), MEAD (Wang et al., 2020), and EARS (Richter et al., 2024), covering a broad range of speakers, acoustic conditions, and conversational styles. As summarized in Table 1, EmotionCoT-35K provides not only wider acoustic granularity but also uniqueness in emotion-centered reasoning. To our knowledge, it is the first prosody-aware CoT dataset tailored for SER, enabling models to produce both emotion label and perceptually grounded explanations.

Table 1: Comparison of different speech captioning datasets across various acoustic features. Reasoning denotes the availability of emotion reasoning with chain-of-thought (CoT) annotations.

| Dataset | Reasoning | Acoustic Feature | | | | | | | | |
|---|---|---|---|---|---|---|---|---|---|---|
| | | Age | Gender | Emotion | Pitch | Speed | Energy | Style | Contour | Stress |
| PromptSpeech (Guo et al., 2023) | ✗ | ✓ | ✓ | ✓ | ✓ | ✓ | ✓ | ✗ | ✗ | ✗ |
| Expresso (Nguyen et al., 2023) | ✗ | ✗ | ✓ | ✓ | ✗ | ✗ | ✗ | ✗ | ✗ | ✗ |
| EARS (Richter et al., 2024) | ✗ | ✓ | ✓ | ✓ | ✓ | ✓ | ✓ | ✓ | ✗ | ✗ |
| SpeechCraft (Jin et al., 2024) | ✗ | ✓ | ✗ | ✓ | ✓ | ✓ | ✓ | ✓ | ✗ | ✓ |
| TextrolSpeech (Ji et al., 2024b) | ✗ | ✗ | ✓ | ✓ | ✓ | ✓ | ✓ | ✗ | ✗ | ✗ |
| CapSpeech (Wang et al., 2025b) | ✗ | ✓ | ✓ | ✓ | ✓ | ✓ | ✓ | ✓ | ✗ | ✗ |
| EmotionCoT-35K | ✓ | ✓ | ✓ | ✓ | ✓ | ✓ | ✓ | ✓ | ✓ | ✓ |

**Dataset construction pipeline.** As shown in Figure 2, we develop an automated annotation pipeline for constructing EmotionCoT-35K. In contrast to traditional manual annotation, this approach is cost-efficient and less prone to human bias, particularly for perceptual variables such as pitch, speed, and energy. Specifically, for feature selection, we prioritize acoustic cues most relevant to human emotion reasoning, ensuring their utility for affective inference. We extract low-level features (speed, pitch, energy) using standard speech processing tools, and identify stressed words from transcriptions using WhiStress (Yosha et al., 2025b). From frame-level pitch–energy trajectories, we derive intonation contours and apply Savitzky–Golay smoothing. The smoothed contours are then classified into coarse styles (expressive vs. flat) and fine-grained intonation patterns (rising, falling, rising–falling, falling–rising). In addition, a wav2vec2.0-based classifier (Baevski et al., 2020) provides speaker attributes, including gender and age group. Building on these annotations, we compile them as contextual prompts for GPT-4o to generate step-wise reasoning traces, with outputs formatted in `<think>` and `<answer>` tags. Details of pipeline and dataset examples are provided in the Appendix.

## 3.2 PROSODY-CENTRIC SUPERVISED FINE-TUNING (SFT)

Directly applying the Qwen2.5-Omni-7B backbone for RL is ineffective, as experiments (Table 5) show that it lacks fine-grained prosody perception—an essential prerequisite for reliable emotion reasoning. For example, humans often rely on prosodic cues such as pitch, energy, and rhythm to infer emotions from speech. An angry utterance is typically characterized by high energy and sharp intonation, whereas a sad one tends to exhibit low pitch and slow speaking rate. These prosodic patterns serve as important perceptual signals for emotional judgment, especially when lexical content alone is ambiguous or emotionally neutral. Thus we introduce a prosody-centric supervised fine-tuning (SFT) stage to build **EmotionThinker-Base** prior to GRPO-based RL (Shao et al., 2024). The SFT corpus (∼500 hours) integrates (i) word-level stress perception using Stress-17K dataset (Yosha et al., 2025a), (ii) prosodic attribute classification tasks derived from expressive ASR data, where the model learns to categorize different pitch, energy, speed, and intonation levels, (iii) comparative prosodic augmentation tasks, where the same utterance is systematically modified with varied pitch, energy, and speed levels, then concatenated in different sequences (e.g., high pitch → low pitch → medium pitch), and the model is trained to identify the correct ordering patterns, and (iv) 5K EmotionCoT samples to expose the model to basic reasoning patterns for the cold-start training. We jointly optimize the audio encoder, audio adapter, and LLM backbone during SFT. This training stage equips EmotionThinker-Base with strong prosody perception ability and an initial grasp of reasoning

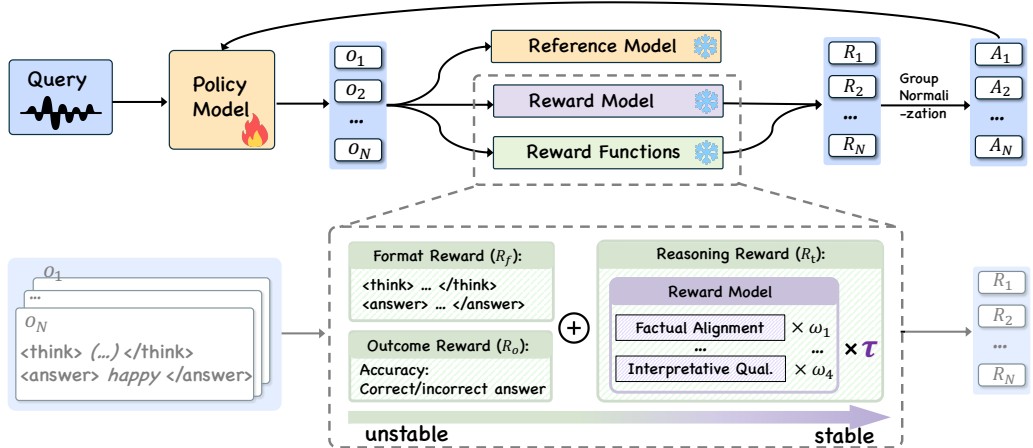

Figure 3: Architecture of EmotionThinker with the proposed GRPO-PTR framework. The upper part depicts the high-level GRPO-PTR training pipeline, where only the policy model is optimized. The lower part details PTR strategy, which progressively introduces reasoning reward to stabilize training and enhance reasoning.

structures, providing a solid foundation for subsequent RL training. EmotionThinker-Base details are provided in the Appendix.

## 3.3 REINFORCEMENT LEARNING FOR EMOTION REASONING

Figure 3 illustrates the RL architecture of EmotionThinker. We follow the GRPO paradigm (Shao et al., 2024) with rule-based rewards. To further supervise reasoning quality, we train a reward model and propose the GRPO-PTR, which progressively guides the reasoning process.

### 3.3.1 RULE-BASED REWARDS

**Format reward.** The format reward, denoted as $R_\mathrm{f}$, enforces adherence to a predefined XML-style schema. Specifically, the reasoning content must be enclosed within `<think>...</think>` and the final prediction within `<answer>...</answer>`. Formally, for an output $o$,

$$R_\mathrm{f}(o) = \begin{cases} 1, & o \text{ follows the format schema,} \\ 0, & \text{otherwise.} \end{cases}$$

**Outcome accuracy reward.** The outcome reward ($R_\mathrm{o}$) verifies whether the predicted emotion label matches the ground truth. For a given instance with gold label $y^*$ and model prediction $\hat{y}$ extracted from `<answer>`, the reward is defined as:

$$R_\mathrm{o}(\hat{y}, y^*) = \begin{cases} 1, & \hat{y} = y^*, \\ 0, & \text{otherwise.} \end{cases}$$

### 3.3.2 GRPO-PTR: PROGRESSIVE TRUST-AWARE REASONING REWARD

Standard GRPO only supervises the final outcomes without constraining the intermediate reasoning process. While in emotion reasoning, this paradigm is prone to shortcut reward hacking: models may generate superficially plausible but logically unsound or perceptually ungrounded reasoning as long as the final answer is correct. To address this limitation, we propose GRPO-PTR (Progressive Trust-aware Reasoning Reward), which augments GRPO with open-ended reasoning supervision to guide the model toward accurate and coherent outputs.

To achieve this, we first train a reasoning reward model that evaluates the quality of open-ended reasoning across multiple dimensions. The model is built upon Qwen2.5-Omni-3B (Xu et al., 2025b) and fine-tuned on 101,400 $(q, r, g)$ tuples, where $q$ is an emotional prompt, $r$ a model-generated reasoning trace, and $g \in \{1, \ldots, 5\}^4$ a vector of criterion-specific scores. The training data are constructed by extending EMOTIONCOT-35K with GPT-4o–synthesized reasoning variants at

controlled quality levels along each criterion. Details are provided in the Appendix. During inference, the reward model outputs a JSON object with four fields: *factual_alignment*, *interpretative_quality*, *caption_completeness*, and *fluency_and_structural_clarity*, each rated on a 1–5 scale. These ratings are then normalized and aggregated into a scalar reasoning reward $R_t$ as follows:

$$R_t = \sum_{j=1}^{4} w_j \, \tilde{g}_j, \qquad \text{where } \tilde{g}_j = \frac{g_j}{5}, \ \sum_j w_j = 1, \ w_j \geq 0.$$

Moreover, since $R_t$ is only a proxy of the unobserved reasoning quality and not directly tied to task correctness $Y \in \{0, 1\}$, it can be unreliable when incorrect responses receive higher $R_t$ scores while correct ones may appear low within a sampled response group. Directly combining such noisy rewards with $R_o$ risks biasing the policy toward spurious reasoning patterns. We addresses this issue by assigning a trustworthiness weight $\tau$ that modulates the contribution of $R_t$ based on its alignment with outcome signals. Specifically, let $R_o^i \in \{0, 1\}$ denote the correctness label of the $i$-th response, where $R_o^i = 1$ indicates the correct prediction and $R_o^i = 0$ indicates the incorrect one. We partition the responses into two disjoint sets:

$$\mathcal{G}_{\text{correct}} = \{i \mid R_o^i = 1\}, \qquad \mathcal{G}_{\text{wrong}} = \{i \mid R_o^i = 0\}.$$

We compute the mean reasoning reward in each group,

$$\bar{R}_t^{(c)} = \frac{1}{|\mathcal{G}_{\text{correct}}|} \sum_{i \in \mathcal{G}_{\text{correct}}} R_t^i, \qquad \bar{R}_t^{(w)} = \frac{1}{|\mathcal{G}_{\text{wrong}}|} \sum_{i \in \mathcal{G}_{\text{wrong}}} R_t^i,$$

and define

$$\tau = \begin{cases} 1, & \bar{R}_t^{(c)} \geq \bar{R}_t^{(w)}, \\ \exp(\bar{R}_t^{(c)} - \bar{R}_t^{(w)}), & \bar{R}_t^{(c)} < \bar{R}_t^{(w)}. \end{cases}$$

This weight $\tau$ shrinks the influence of $R_t$ when its distribution does not separate correct from incorrect responses (i.e., when $\bar{R}_t^{(c)} < \bar{R}_t^{(w)}$), reducing the risk of reinforcing misleading reasoning trajectories. In essence, $\tau$ serves as a group-level alignment gate: only when the sampled response set exhibits global agreement between reasoning quality and outcome correctness does $R_t$ contribute to the reward signal.

The overall reward for GRPO-PTR optimization as follows:

$$R_i = \alpha_f R_f + \alpha_o R_o + \alpha_t \tau \cdot R_t$$

where $\alpha$ denotes the weight of each reward component. To ensure stable optimization, we adopt a progressive reward scheduling strategy: during the early training stage, we optimize solely with the outcome reward $R_o$ and format reward $R_f$ until emotion accuracy consistently achieves certain level (e.g., 50% accuracy). Introducing multiple unstable rewards too early can cause random fluctuations in any component to dominate the reward sum and produce incorrect advantage signals, thereby impeding model convergence. Since the reasoning reward $R_t$ is particularly challenging to optimize due to the diversity of open-ended generation, we delay its incorporation until the model has stably adapted to the rule-based rewards. This progressive strategy effectively mitigates early-stage instability and ensures more stable reward optimization.

## 4 EXPERIMENT

### 4.1 EXPERIMENTAL SETUP

**Datasets and evaluation.** The RL post-training is conducted on 30K samples from EmotionCoT-35K training data. For evaluation, we adopt four widely used SER benchmarks: IEMOCAP (Busso et al., 2008), MELD (Poria et al., 2019), RAVDESS (Livingstone & Russo, 2018), and SAVEE (Jackson & Haq, 2014). Among them, we use the IEMOCAP and MELD test sets, while RAVDESS and SAVEE are evaluated in a zero-shot setting.

**Baselines.** To comprehensively benchmark the emotion reasoning ability of EmotionThinker, we compare against 13 general-purpose SpeechLLMs and OmniLLMs (GLM-4-Voice (Zeng et al.,

Table 2: Performance comparison across models. Emotion recognition is measured by accuracy (%), while reasoning quality is assessed on the overall test dataset on four dimensions: Factual Alignment (FA.), Interpretative Quality (IQ.), Caption Completeness (CC.), and Fluency and Structure (FS.), each on a 5-point scale. Top two results are highlighted in **bold** and underline, respectively.

| Models | Emotion Recognition Accuracy ↑ | | | | | Emotion Reasoning ↑ | | | | |
|---|---|---|---|---|---|---|---|---|---|---|
| | IEMOCAP | MELD | RADESS | SAVEE | Avg | FA. | IQ. | CC. | FS. | Avg |
| *General Speech Large Language Models* | | | | | | | | | | |
| GLM-4-Voice | 22.38 | 21.43 | 19.67 | 20.84 | 21.29 | 1.95 | 2.62 | 2.17 | 3.22 | 2.49 |
| BLSP | 41.24 | 50.47 | 11.10 | 10.77 | 36.02 | 1.01 | 2.44 | 1.63 | 2.71 | 1.94 |
| DIVA | 40.15 | 35.19 | 18.28 | 20.14 | 31.87 | 2.33 | 2.96 | 2.68 | 3.65 | 2.90 |
| MERaLiON | 45.66 | 37.98 | 12.43 | 16.25 | 33.09 | 1.17 | 2.03 | 1.88 | 3.54 | 2.16 |
| MERaLiON2 | 51.05 | 51.10 | 37.02 | 25.43 | 46.09 | 2.41 | 3.20 | 2.72 | 3.83 | 3.04 |
| SALMONN | 23.82 | 31.32 | 20.38 | 17.86 | 25.61 | 1.32 | 2.11 | 1.03 | 2.89 | 1.84 |
| Qwen-Audio-Chat | 38.80 | 55.70 | 70.11 | 71.53 | 54.86 | 1.98 | 2.40 | 2.07 | 2.79 | 2.31 |
| Qwen2-Audio-Instruct | 37.71 | 51.23 | 64.98 | 65.13 | 51.14 | 1.70 | 2.27 | 1.90 | 2.85 | 2.18 |
| Kimi-Audio | 57.72 | 59.13 | 61.07 | 55.21 | 58.83 | 2.45 | 2.75 | 2.35 | 3.34 | 2.72 |
| *Emotion Focused Speech Large Language Models* | | | | | | | | | | |
| SECap | 36.29 | 34.20 | 28.34 | 21.93 | 32.64 | 1.82 | 1.13 | 1.00 | 2.31 | 1.56 |
| OSUM-EChat | 41.49 | 53.38 | 37.34 | 24.27 | 44.04 | 2.18 | 2.59 | 2.21 | 3.19 | 2.54 |
| BLSP-Emo | 76.00 | 57.30 | **72.00** | 63.73 | 65.41 | 2.33 | 2.78 | 2.45 | 3.37 | 2.73 |
| *Omni Large Language Models* | | | | | | | | | | |
| Phi-4-Multimodal | 36.62 | 39.81 | 17.63 | 15.55 | 32.15 | 2.12 | 2.67 | 2.51 | 3.43 | 2.69 |
| Megrez-3B-Omni | 14.77 | 21.89 | 21.45 | 16.97 | 19.25 | 2.10 | 2.56 | 2.41 | 3.33 | 2.61 |
| MiniCPM-O | 35.54 | 52.78 | 40.93 | 35.47 | 43.60 | 2.53 | 3.00 | 2.71 | 3.81 | 3.01 |
| Qwen2.5-Omni-7B | 45.70 | 54.64 | 64.77 | 52.49 | 50.83 | 2.32 | 2.91 | 2.64 | 3.59 | 2.87 |
| EmotionThinker | **77.68** | **59.71** | 71.56 | **73.96** | **68.89** | **3.54** | **4.01** | **3.96** | **4.42** | **3.98** |

2024), BLSP (Wang et al., 2023), DIVA (Held et al., 2024), MERaLiON (He et al., 2025), MERaLiON2 (He et al., 2025), Qwen-Audio-Chat (Chu et al., 2023), Qwen2-Audio-Instruct (Chu et al., 2024), Kimi-Audio (KimiTeam et al., 2025), Phi-4-Multimodal (Abouelenin et al., 2025), Megrez-3B-Omni (Infinigence AI, 2024), MiniCPM-O (MiniCPM-o Team, 2024), Qwen2.5-Omni-7B (Xu et al., 2025b)) and 3 emotion-focused SpeechLLMs (BLSP-Emo (Wang et al., 2024), SECap (Xu et al., 2024), OSUM-EChat (Geng et al., 2025)).

**Implementation details.** EmotionThinker is initialized from EmotionThinker-Base, which is trained for one epoch on 500+ hours of prosody-enhanced speech data. RL post-training runs for 3,000 steps with a KL divergence coefficient of 0.04, a learning rate of $1 \times 10^{-6}$, and $K = 8$ sampled candidates per input. The overall reward is a weighted sum of accuracy, format, and reasoning signals with weights 1.0, 0.3, and 0.5, respectively.

## 4.2 MAIN RESULTS

We assess model performance from two perspectives: unweighted accuracy for emotion recognition and GPT-based ratings on a 5-point scale for reasoning quality. The reasoning evaluation covers four dimensions: (1) Factual Alignment (FA): whether the explanation accurately reflects speaker traits, semantic content, and prosodic cues in the audio; (2) Interpretative Quality (IQ): how clearly and convincingly these cues are connected to the emotion outcome; (3) Caption Completeness (CC): the degree to which the

Table 3: Human evaluation results on emotion reasoning based on a consistent 100-sample set.

| Models | FA. | IQ. | CC. | FS | Avg |
|---|---|---|---|---|---|
| BLSP-Emo | 1.2 | 2.0 | 1.5 | 2.5 | 1.8 |
| OSUM-EChat | 1.5 | 2.5 | 1.3 | 2.5 | 2.0 |
| Kimi-Audio | 2.6 | 3.4 | 1.6 | 3.5 | 2.8 |
| Qwen2.5-Omni | 2.4 | 4.2 | 3.0 | 4.5 | 3.5 |
| EmotionThinker | **3.7** | **4.5** | **4.7** | **4.5** | **4.4** |

caption covers all relevant acoustic and semantic information in a comprehensive manner; and (4) Fluency and Structural Clarity (FS): measures the grammatical correctness, readability, and logical organization of the reasoning. Detailed prompt and rubric are provided in the Appendix.

**Qwen2.5-Omni-7B:**
<think> The emotion expressed in the audio clip is 'angry'. This is indicated by the tone and intensity of the speaker's voice, which suggests frustration or irritation.
</think><answer>angry</answer>

**Question:**
**What is the emotion expressed in this audio clip? (sad)**

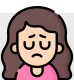

**EmotionThinker-Base + GRPO:**
<think> The speaker's tone suggests a state of sad. The energy level is relatively steady, and while not particularly low, it could imply a lack of strong emotion…\n\nThe transcription—"Okay, I'm sorry, but I just stood in this line for an hour"—reveals a context in which the speaker is apologizing for arriving late, possibly due to being delayed by something exciting or important. The use of the word "sorry" seems more sarcastic than sincere, especially when paired with an upward intonation pattern that hints at playful deflection…Overall, the speech likely reflects playful annoyance or energetic sarcasm, rather than any negative emotion such as frustration or sadness…</think><answer>sad</answer>

**EmotionThinker (GRPO-PTR):**
<think> The speaker, a teenage female, conveys a sense of emotional tension through several prosodic and acoustic cues. The pitch level is notably low, which often signals a subdued or restrained emotional state, potentially indicative of sadness. This is further supported by the low energy level, which suggests the speaker is not exerting much vocal effort, a common trait when someone feels drained or exasperated.\n\nThe speech rate is normal, which rules out heightened urgency or panic, but the stress placed on the word \"hour\" stands out. By emphasizing this word, the speaker draws attention to the duration of their experience, which likely feels burdensome or unfair to them. This stress pattern aligns with the semantic content of the transcription, where the speaker mentions standing in a line for an extended period: \"I just stood in this line for an hour.\" The phrasing itself hints at dissatisfaction or impatience, reinforcing the emotional tone… the overall delivery—marked by low pitch and energy—suggests a lack of enthusiasm or patience. Taken together, these prosodic features, combined with the semantic context, strongly point to an emotional state of sad or frustrated.</think><answer>sad</answer>

Figure 4: Case study comparing emotional reasoning across models on a sad utterance. Red highlights mark descriptions inconsistent with the ground truth, whereas green highlights indicate correct evidence.

As shown in Table 2, EmotionThinker achieves state-of-the-art performance on both overall emotion recognition accuracy and reasoning quality. For emotion recognition, it demonstrates superior accuracy on most benchmarks, achieving an overall average of 68.89%. This surpasses the second-best model, BLSP-Emo, by approximately 3%. Notably, most general-purpose models (e.g., Kimi-Audio) outperform the other two emotion-focused SpeechLLMs (SECap and OSUM-EChat), underscoring the need for further advances in emotion-specialized SpeechLLMs. For reasoning quality, most baselines lack native structured reasoning, except Qwen2.5-Omni with <think><answer> formatting. To elicit their latent reasoning ability, we apply a unified prompt that requests justifications grounded in speaker traits, prosodic cues, and semantic content. EmotionThinker achieves the highest average score (3.98), substantially outperforming all models across reasoning dimensions. Example reasoning outputs per model are included in the Appendix.

To mitigate potential biases in GPT-based evaluations, we conduct additional human assessment on five representative models using 100 anonymized outputs (20 per model), which are rated by four independent reviewers. As shown in Table 3, EmotionThinker demonstrates clear superiority across all dimensions, with particularly stronger performance in Factual Alignment (3.7) and Caption Completeness (4.7). The model performance rankings align closely with GPT-based evaluations, indicating robust agreement between human and automatic judgments.

## 4.3 ANALYSIS

To validate the effectiveness of our EmotionThinker framework, we conduct a comprehensive analysis. We first examine the prosody perception capabilities of EmotionThinker-Base. Second, we compare supervised fine-tuning and reinforcement learning under the same training data to assess the benefits of reward-driven optimization. Third, we perform component-wise ablations to examine the contribution of core components in our proposed GRPO-PTR framework, accompanied by a case study. Finally, we analyze the impact of key hyperparameters, including sampling size and reward weighting.

Table 4: Ablation study on different training strategies for average speech emotion recognition accuracy (SER) and emotion reasoning (ER) score, evaluated on the overall test dataset. Variants V1–V6 are built upon Baseline 2.

| Variations | Training Strategy | SER | ER |
|---|---|---|---|
| | | (Acc)↑ | (Score)↑ |
| Baseline 1 | Qwen2.5-Omni-7B | 50.83 | 2.87 |
| Baseline 2 | EmotionThinker-Base | 52.63 | 3.41 |
| V1 | SFT | 53.91 | 3.78 |
| V2 | GRPO | 62.91 | 3.45 |
| V3 | GRPO-PTR-w/o-trained-RM | 66.67 | 3.36 |
| V4 | GRPO-PTR-w/o-trust | 67.71 | 3.74 |
| V5 | GRPO-PTR-w-progressive | 62.80 | 3.76 |
| V6 | GRPO-PTR | **68.89** | **3.98** |

**Prosody perception comparison.** We construct a prosody perception test set by sampling 1,000 unseen utterances from the prosody-centric dataset. As shown in Table 5, EmotionThinker-Base demonstrates significantly improved sensitivity to pitch, energy, speed, and stress compared to the original backbone. These results validate the necessity and effectiveness of prosody-centric SFT for enhancing prosodic understanding before RL.

Table 5: Prosody perception comparison across pitch (Pit.), speed (Spee.), energy (Ene.), intonation (Into.), and stress (Stre.). Evaluation is based on accuracy (%).

| Model | Pit. | Spee. | Ene. | Into. | Stre. |
|---|---|---|---|---|---|
| Qwen2.5-Omni-7B | 25.71 | 29.94 | 27.67 | 25.83 | 30.24 |
| EmotionThinker-Base | **75.11** | **68.70** | **69.42** | **60.25** | **71.50** |

Table 6: GRPO-PTR with varying K settings.

| Settings | Iemocap | Meld | Radess | Savee | Avg |
|---|---|---|---|---|---|
| K = 4 | 75.61 | 57.64 | 70.29 | 72.85 | 67.06 |
| K = 6 | 75.31 | 57.55 | **71.01** | 72.55 | 67.07 |
| K = 8 | **75.82** | 58.03 | 70.58 | **72.91** | 67.35 |
| K = 10 | 75.77 | 57.89 | 70.59 | 72.87 | 67.28 |
| K = 16 | 75.81 | **59.22** | 69.87 | 72.85 | **67.66** |

**Supervised fine-tuning (SFT) vs. reinforcement learning (RL).** We compare the SFT with different RL approaches using the same training data from EmotionCoT-35K. As shown in Table 4, SFT (V1) provides moderate gains over the base model, achieving 53.91% accuracy and 3.78 in reasoning score. Standard GRPO (V2), which relies only on rule-based rewards, yields a clear improvement. Our proposed GRPO-PTR (V6) further enhances both emotion accuracy and reasoning quality.

As shown in Table 4, we ablate key components of GRPO-PTR: (i) GRPO-PTR-w/o-trained RM, where the reasoning reward is derived from an untrained Qwen2.5-Omni-3B model; (ii) GRPO-PTR-w/o-trust, where the trustworthiness weight $\tau$ is removed; and (iii) GRPO-PTR-w/o-progressive, where progressive reward combination is disabled and all reward signals are aggregated simultaneously. **Effectiveness of the reward model**: When the reasoning reward is derived from an untrained model (V3), both SER accuracy and ER score drop notably (66.67% and 3.36, respectively). This indicates that our trained reward model provides more informative supervision signals, whereas the untrained model introduces noise that weakens both task correctness and reasoning quality. **Effect of the trustworthiness weight**: Removing the trustworthiness weight $\tau$ (V4) leads to only a marginal decline in SER accuracy (67.71%) but causes a notable drop in ER score (3.74). This suggests that $\tau$ helps filter out unreliable yet seemingly plausible reasoning. **Effectiveness of progressive reward combination**: Disabling progressive scheduling (V5) results in a notable performance drop, with SER accuracy decreasing to 62.80% and ER score to 3.76. This shows the necessity of integrating a progressive reward integration strategy to improve training stability.

**Case study.** As illustrated in Figure 4, compared with Qwen2.5-Omni-7B and EmotionThinker-Base + GRPO, GRPO-PTR produces more accurate and comprehensive prosody-grounded reasoning. It demonstrates stronger alignment with acoustic and semantic cues, yielding more coherent explanations. In contrast, Qwen2.5-Omni-7B provides overly brief reasoning without sufficient justification, while the GRPO variant includes hallucinated interpretations inconsistent with the input.

**Ablations on hyperparameter settings.** As shown in Table 6, we vary the number of generated responses K, while keeping all other settings fixed during training. Table 7 shows that reducing K from 16 to 4 has only a marginal effect, we choose 8 for trade-off between computational cost and accuracy. In addition, we conduct a sensitivity analysis on two parameters $\alpha_o$ and $\alpha_t$, which

Table 7: Sensitivity analysis on reward penalty.

| Settings | Iemocap | Meld | Radess | Savee | Avg |
|---|---|---|---|---|---|
| $\alpha_o$=1.0, $\alpha_t$=0.3 | **77.71** | 59.68 | 71.55 | 73.92 | 68.87 |
| $\alpha_o$=1.0, $\alpha_t$=0.5 | 77.68 | **59.71** | **71.56** | **73.96** | **68.89** |
| $\alpha_o$=1.0, $\alpha_t$=1.0 | 73.60 | 57.46 | 68.41 | 66.92 | 65.52 |

correspond to the outcome accuracy reward and reasoning reward weights. We fix the format reward weight $\alpha_f$ as 0.3 for all settings. Results suggest that moderate emphasis on thinking reward ($\alpha_t = 0.5$) yields the best overall performance. A higher $\alpha_t$ ($\alpha_t = 1.0$) leads to slight performance degradation, likely due to optimization instability introduced by excessive emphasis on intermediate reasoning signals. These results emphasize the importance of maintaining a balanced reward composition in multi-signal reinforcement learning.

## 5 CONCLUSION

In this work, we present EmotionThinker, a pioneering RL-based framework that extends speech emotion recognition (SER) from simple classification to explainable emotion reasoning. Within the RL stage, we proposed the GRPO-PTR strategy to refine reasoning in a coherent and trustworthy manner. This design enables the model to move beyond surface-level label prediction, yielding explanations that are fine-grained, logically coherent, and grounded in relevant acoustic and semantic cues. Comprehensive experiments on various benchmarks show that EmotionThinker achieves state-of-the-art performance in both emotion accuracy and the reasoning quality.

## 6 ACKNOWLEDGEMENT

This work was supported by the Centre for Perceptual and Interactive Intelligence (CPII) Ltd., a CUHK-led InnoCentre under the InnoHK initiative of the Innovation and Technology Commission of the Hong Kong Special Administrative Region Government.

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

# EmotionThinker: Prosody-Aware Reinforcement Learning for Explainable Speech Emotion Reasoning

## *Supplementary Material*

CONTENTS

## A  THE USE OF LARGE LANGUAGE MODELS

During this work, we used large language models (LLMs) as assistive tools for data construction, model evaluation, and paper polishing. For data construction, we used the GPT-4o API to synthesize samples for EmotionCoT-35K and to create training pairs for the reward model (see Section B and Section D); For evaluation, GPT-4o was used to assess the quality of emotion reasoning under a fixed rubric (see Section E). For writing, LLMs were used only for copyediting and phrasing to improve clarity and consistency.

## B  EMOTIONCOT-35K

### B.1  DATA SOURCE

**IEMOCAP (Interactive Emotional Dyadic Motion Capture) (Busso et al., 2008):** The IEMOCAP database is an acted, multimodal and multispeaker database, recently collected at SAIL lab at USC. It contains approximately 12 hours of audiovisual data, including video, speech, motion capture of face, text transcriptions. It consists of dyadic sessions where actors perform improvisations or scripted scenarios, specifically selected to elicit emotional expressions. IEMOCAP database is annotated by multiple annotators into categorical labels, such as anger, happiness, sadness, neutrality, as well as dimensional labels such as valence, activation and dominance.

**MELD (Multimodal EmotionLines Dataset) (Poria et al., 2019):** The MELD dataset has been created by enhancing and extending EmotionLines dataset. MELD contains the same dialogue instances available in EmotionLines, but it also encompasses audio and visual modality along with text. MELD has more than 1400 dialogues and 13000 utterances from Friends TV series. Multiple speakers participated in the dialogues. Each utterance in a dialogue has been labeled by any of these seven emotions – Anger, Disgust, Sadness, Joy, Neutral, Surprise and Fear. MELD also has sentiment (positive, negative and neutral) annotation for each utterance.

**Expresso (Nguyen et al., 2023):** Expresso, a high-quality (48kHz) expressive speech dataset that includes both expressively rendered read speech (8 styles, in mono wav format) and improvised dialogues (26 styles, in stereo wav format). The dataset includes 4 speakers (2 males, 2 females), and totals 40 hours (11h read, 30h improvised).

**MEAD (Multi-view Emotional Audio-visual Dataset) (Wang et al., 2020):** MEAD is a large-scale audio-visual dataset built for research in emotional talking-face generation and cross-modal expression modeling. It includes data from 60 actors and actresses, recorded under eight distinct emotions at three intensity levels (except neutral) in a strictly controlled environment.

**EARS (Expressive Anechoic Recordings of Speech) (Richter et al., 2024):** It consists of ~100 hours of speech from 107 speakers of diverse demographics (age, ethnicity, background), recorded at 48 kHz in anechoic conditions. EARS covers a broad range of expressive styles: multiple reading styles (e.g. regular, loud, whisper, fast, slow, high pitch, low pitch), non-verbal vocalizations, conversational freeform speech, and speech in 22 emotional styles. Transcriptions are provided for the reading portion, along with metadata (gender, age, race, native language). Its high quality, style diversity, and speaker variation make it particularly suitable for modeling expressive prosody, style transfer, and robust speech enhancement.

### B.2  AUTOMATIC ANNOTATION PIPELINE

To enable large-scale, fine-grained expressive speech annotation, this work introduces a modular automatic annotation pipeline that systematically extracts multiple prosodic and speaker-level attributes—including pitch, energy, speaking rate, word-level stress, gender, and age—using a combination of signal processing techniques and specialized classification models. The core prosodic features, namely pitch, energy, and speed, are derived via classical digital signal processing methods:

pitch and energy are computed from short-time acoustic frames, while speaking rate is estimated by combining phoneme-level alignments from forced alignment tools with utterance duration statistics. For stress annotation, we employ the WhiStress model (Yosha et al., 2025b)—an alignment-free framework that extends a frozen Whisper backbone with a stress detection head comprising a transformer decoder block and a feedforward classifier. It predicts token-level stress labels based on cross-attention over intermediate Whisper embeddings, achieving high accuracy without requiring forced alignment or manual labels. For intonation contour annotation, we first extract frame-level pitch and energy trajectories, then apply a Savitzky–Golay filter to smooth the curves and reduce local noise. The smoothed pitch contours are analyzed to derive both coarse-grained intonational styles (e.g., expressive vs. flat) and fine-grained melodic patterns (e.g., rising, falling, rising–falling, falling–rising), which reflect the global prosodic dynamics of the utterance. To capture speaker characteristics, gender and age are inferred using supervised classification models trained atop wav2vec 2.0 (Baevski et al., 2020) representations. Specifically, gender prediction achieves high accuracy by distinguishing between male and female speakers, while age estimation bins speakers into coarse-grained age categories. These automatically extracted attributes serve as the foundation for natural language description generation, and their accuracy is independently evaluated on external benchmarks, demonstrating reliability across both English and Mandarin domains.

### B.3 CoT Prompt

As shown in Figure 5, we use a GPT-4o prompt to construct emotion reasoning process. The prompt guides the model to generate structured explanations grounded in both prosodic and semantic cues, enabling CoT-style supervision.

## C EmotionThinker-Base

### C.1 Details of the Prosody-Centric SFT Corpus

**(i) Word-level stress dataset (Stress-17K (Yosha et al., 2025a)).** Stress-17K is generated through a synthetic pipeline that varies sentence stress placement to alter utterance meaning, providing paired audio samples with explicit stress annotations. It covers multiple stress categories (contrastive, emphatic, new-information, focus) and offers both detection and reasoning supervision. We directly incorporate this dataset to enhance the model's capacity for explicit word-level stress perception.

**(ii) Prosodic Attribute Classification Data.** We sample utterances from the GigaSpeech (Chen et al., 2021) corpus and process them using the annotation pipeline originally developed for constructing EmotionCoT. This pipeline extracts pitch, energy, speaking rate (speed), and intonation information. Based on these annotations, we construct supervised fine-tuning (SFT) tasks where the model is queried about prosodic categories. For example, given an audio clip, the corresponding question explicitly asks "What is the pitch level of this speech?". In this way, the model learns to associate acoustic signals with categorical prosodic attributes.

**(iii) Comparative prosodic augmentations.** To strengthen robustness, we construct prosodic contrastive data through signal-level augmentations applied to GigaSpeech (Chen et al., 2021) utterances. Specifically, we generate modified versions of each sample with systematically increased or decreased pitch, energy, and speaking rate. These variations are then randomly concatenated to create sequences exhibiting controlled prosodic shifts. For instance, a concatenated sequence may follow a progression such as low → high → medium energy. Corresponding training questions explicitly ask the model to identify the prosodic ordering pattern. This design encourages the model to perform comparative reasoning over prosodic attributes rather than relying on absolute values alone.

### C.2 Training Details of EmotionThinker-Base

The backbone of EmotionThinker-Base is Qwen2.5-Omni-7B (Xu et al., 2025b). The supervised fine-tuning (SFT) process is divided into two stages. For Stage I: we train on the constructed *Prosody-Centric corpus*, jointly updating the audio encoder, audio adapter, and LLM backbone with full-parameter training. To preserve the backbone's instruction-following ability and basic

---

## Prompt Template (Generate emotion reasoning process)

You are an expert in emotional speech and vocal analysis. You have listened to an audio sample with the following cues:

- `groundtruth_emotion`: "{item['gt']}"
- `transcription`: "{transcription}"
- `speaker_gender`: "{item['gender_new']}"
- `speaker_age`: "{item['age_level']}"
- `pitch_level`: "{item['pitch_level']}"
- `energy_level`: "{item['energy_level']}"
- `speed_level`: "{item['speed_level']}"
- `intonation_contour`: "{item['intonation_contour']}"
- `stressed_word`: "{item['stress']}"

Your task is to provide a detailed reasoning explanation of **why** the speaker is likely expressing the emotion labeled as `{item['gt']}`.

**Important:** Assume you do **not** know the ground-truth label. Like a human listener, infer the most likely emotion through step-by-step reasoning. Let the conclusion arise naturally from your reasoning process. The reasoning process focus should be on **acoustic and prosodic cues**, while semantic content may serve as a secondary reference.**

**Requirements:**
- Explicitly reference and quoting the **transcription** in your reasoning
- Naturally incorporate the speaker profile into your writing (e.g., "a middle-aged female speaker").
- Discuss key acoustic/prosodic features you hear, such as pitch, energy, speech rate, intonation, stress, etc. Focus on features that you believe strongly support the predicted emotion, and you may omit or downplay others that seem unimportant or irrelevant.
- Aim for stylistic variation across outputs by using diverse sentence structures and distributing your reasoning across multiple, well-structured sentences.
- When analyzing the semantic content, consider whether it aligns with the labeled emotion. If it does, point out specific cues (e.g., frustration, concern); if not, explain how the emotion may still be conveyed prosodically despite the content.
- If the original given `intonation_contour` is "uncertain" or "too_short", you can ignore this part analysis.
- You may briefly mention stress patterns if they provide insight, but you are encouraged to omit details that seem incidental or not emotionally meaningful.
- Length: 50 – 200 words. Do **not** exceed 200 words.
- Do **not** begin your explanation with the emotion label. Instead, simulate a human reasoning process where the emotional interpretation emerges gradually through observation and analysis.

Only return the emotional reasoning content.

Figure 5: Prompt template used to elicit emotion reasoning traces from GPT-4o.

ASR capability, we additionally incorporate 20% text-only data and 20% ASR data sampled from LibriSpeech (Panayotov et al., 2015) and GigaSpeech (Chen et al., 2021). Training is conducted for one epoch with a learning rate of $1 \times 10^{-5}$. For Stage II: we further sample 5K examples from *EmotionCoT* for cold-start reasoning supervision. In this stage, only the LLM layers are trained with LoRA (Hu et al., 2022) adaptation, while the audio encoder and adapter remain fixed. Training is performed for two epochs with a learning rate of $1 \times 10^{-5}$.

# D  REWARD MODEL

## D.1  DATA CONSTRUCTION DETAILS

For constructing the training data of the thinking reward model, we build upon the EmotionCoT corpus. We first sample 20K high-quality instances from EmotionCoT, treating them as gold-standard reasoning traces with perfect scores (5) across the four evaluation dimensions: "Factual Alignment", "Interpretative Quality", "Caption Completeness", and "Fluency and Structural Clarity". To introduce controlled quality variation, we create degraded counterparts by randomly assigning a score from 1 to 5 for each dimension. This procedure produces 101,400 $(q, r, g)$ tuples in total, where $q$ denotes the emotional prompt, $r$ the reasoning variant, and $g$ denotes the criterion-wise score vector, where each element is an integer from 1 to 5. Given the assigned score configuration, GPT-4o is prompted to generate reasoning traces that reflect the intended quality levels. For example, lowering the score on "Factual Alignment" leads to factually inconsistent content, while a lower "Fluency and Structural Clarity" score yields disfluent or poorly structured text. This systematic process ensures balanced coverage across quality levels and provides diverse supervision signals for training the reward model.

## D.2  GPT PROMPT

To construct reasoning responses, we use GPT-4o with the prompt template shown in Figure 6 and Figure 7. Given a target score configuration across the four dimensions, GPT-4o generates reasoning text that matches the specified quality levels, ensuring controlled variation for the reward model supervision.

# E  EMOTION REASONING CRITERIA

We obtain 1–5 ratings on four key dimensions of emotion reasoning quality using GPT-4o: (1) *Factual Alignment*, (2) *Interpretative Quality*, (3) *Caption Completeness*, and (4) *Fluency and Structural Clarity*. The scoring prompt used to elicit these criterion-wise ratings is shown in Figure 8 and Figure 9.

# F  CASE STUDY

To better understand how different models perform emotion reasoning, we present a detailed case study comparing their output on the same speech input. The selected example corresponds to a female speaker expressing the line: "Okay, I'm sorry, but I just stood in this line for an hour. Can I—is there any way I can— " The ground truth emotion label for this utterance is **sad**.

As shown in Table 8, we evaluate and compare the reasoning outputs of **EmotionThinker** with 11 other representative SpeechLLMs in this example. The models include **Kimi-Audio** (KimiTeam et al., 2025), **Qwen2-Audio-Instruct** (Chu et al., 2024), **Qwen2.5-Omni-7B** (Xu et al., 2025b), **BLSO-Emo** (Wang et al., 2024), **Phi-4-Multimodal** (Abouelenin et al., 2025), **MERaLiON2** (He et al., 2025), **DIVA** (Held et al., 2024), **Megrez-3B-Omni** (Infinigence AI, 2024), **MiniCPM-O** (MiniCPM-o Team, 2024), **GLM-4-Voice** (?), and **OSUM-EChat** (Geng et al., 2025). The case study reveals that **EmotionThinker** demonstrates more accurate and comprehensive capture of acoustic information, along with stronger logical consistency in emotion reasoning.

## Prompt Template (Generate reward model training data )

You are an expert in emotional speech and vocal analysis. The given audio sample with the following cues:

- `groundtruth_emotion`: "{item['major_emotion']}"
- `transcription`: "{transcription}"
- `speaker_gender`: "{item['gender_new']}"
- `speaker_age`: "{item['age_level']}"
- `pitch_level`: "{item['pitch_level']}"
- `energy_level`: "{item['energy_level']}"
- `speed_level`: "{item['speed_level']}"
- `intonation_contour`: "{item['intonation_contour']}"
- `stressed_word`: "{item['stress']}"

**Suppose you have listened to the audio sample**. Your task is to provide a **target quality level** reasoning explanation of **why** the speaker is likely expressing the emotion labeled as `{item['major_emotion']}`.

---
### When you generate certain quality score reasoning content, you should follow the quality rules below (1–5 score range):

#### 1. **Factual Alignment**
How accurately the explanation incorporates the provided metadata (e.g., pitch, stress, transcription).

- **5**: Perfect use of all relevant metadata with no factual errors. Deep integration of acoustic cues.
- **4**: Mostly accurate usage of metadata with minor omissions or inaccuracies.
- **3**: Partial use of metadata; includes some mistakes or superficial mentions.
- **2**: Sparse and incorrect use of metadata; core cues are missing or misused.
- **1**: Fails to use or misrepresents the provided metadata entirely.

#### 2. **Interpretative Quality**
How logically and persuasively the reasoning connects acoustic and linguistic features to emotional interpretation (regardless of factual correctness).

- **5**: Clear, logical, and compelling reasoning; draws insightful connections between prosody and emotion.
- **4**: Mostly strong and logical interpretation, though some reasoning may be underdeveloped.
- **3**: Reasoning is present but shallow or formulaic; some connections are unclear.
- **2**: Weak interpretative logic; vague or unconvincing justification.
- **1**: No meaningful reasoning; assertion without explanation.

Figure 6: Prompt template for generating reward model training data (Part 1).

Table 8: Case study comparing emotion reasoning outputs from EmotionThinker and 12 representative SpeechLLMs on the same audio sample. The ground-truth label is sad, and the analysis highlights differences in prosodic cue recognition, semantic integration, and logical coherence across models. EmotionThinker demonstrates more accurate and comprehensive capture of acoustic information, together with stronger logical consistency in its emotion reasoning.

| Model | Reasoning |
|---|---|
| Ground Truth | The speaker's tone suggests an emotional state that feels subdued or resigned. The pitch level is notably low, which often conveys a sense of heaviness or dejection, especially in contrast to more dynamic or elevated pitch patterns associated with excitement or anger. This low pitch, combined with the normal energy level, implies the speaker isn't entirely drained but lacks enthusiasm or vigor, a hallmark of someone carrying a weight of sadness or disappointment. The transcription offers further insight. The line "Okay, I'm sorry, but I just stood in this line for an hour" carries an apologetic tone, which is reinforced by the word "sorry." Apologies, when coupled with a low pitch and steady, normal pacing, often signal emotional vulnerability or regret. The stress on the word "hour" suggests the speaker is emphasizing the duration of their wait, likely reflecting a sense of frustration or resignation about wasted time. However, this frustration doesn't come across as angry; instead, the calm pacing and lack of heightened energy suggest a quieter, more reflective emotional state. The speaker's profile as a female teenager might also contribute to the interpretation, as younger individuals in such situations may express disappointment in a less confrontational and more uncertain manner. Overall, the delivery aligns with an emotion rooted in sadness or subdued frustration. |
| EmotionThinker (ours) | The speaker, a teenage female, conveys a sense of emotional tension through several prosodic and acoustic cues. The pitch level is notably low, which often signals a subdued or restrained emotional state, potentially indicative of frustration or resignation. This is further supported by the low energy level, which suggests the speaker is not exerting much vocal effort, a common trait when someone feels drained or exasperated. The speech rate is normal, which rules out heightened urgency or panic, but the stress placed on the word "hour" stands out. By emphasizing this word, the speaker draws attention to the duration of their experience, which likely feels burdensome or unfair to them. This stress pattern aligns with the semantic content of the transcription, where the speaker mentions standing in a line for an extended period: "I just stood in this line for an hour." The phrasing itself hints at dissatisfaction or impatience, reinforcing the emotional tone. While the intonation contour is uncertain, the overall delivery—marked by low pitch and energy—suggests a lack of enthusiasm or patience. Taken together, these prosodic features, combined with the semantic context, strongly point to an emotional state of frustration or irritation. |
| Qwen2-Audio-Instruct (Chu et al., 2024) | Based on the speaker's tone, intonation, and word choice, the speaker seems to be expressing a sense of confusion or perplexity. The use of the word 'can' (to the end) suggests that the speaker is trying to understand or clarify something, but the question mark at the end implies that they are unsure about the answer. Additionally, the speaker's voice carries a light and high-pitched quality, which often corresponds with feelings of confusion or bewilderment. Overall, the combination of these factors leads me to believe that the speaker is feeling confused. |

| Qwen2.5-Omni-7B (Xu et al., 2025b) | The speaker is expressing neutral emotion in this audio clip. Reasoning: 1. Transcription: The speaker says, "Hi, I need an ID." This is a neutral statement, neither positive nor negative, indicating a straightforward request or instruction. 2. Prosody: The tone of the voice is even and lacks emotional inflection. There is no rise or fall in pitch that would suggest excitement, sadness, or anger. 3. Context: The phrase "I need an ID" is a common, everyday request that does not evoke strong emotions. It is a neutral, matter-of-fact statement. 4. Speaker Traits: The speaker's tone is steady and unemotional, which aligns with a neutral emotional state. In summary, the speaker's neutral emotion is evident from the straightforward nature of the request and the even tone of voice. |
|---|---|
| BLSP-EMO (Wang et al., 2024) | The emotion tone of the speech provided is sad. The speaker says they waited in line for an hour, which shows they are unhappy with the situation. Even though they start with "I'm sorry," their tone suggests they are annoyed and want a solution quickly. The incomplete sentence at the end ("can I... is there any way I can— ") shows they are struggling to stay polite while also trying to get help. These signs together reflect sadness. |
| Phi-4-Multimodal (Abouelenin et al., 2025) | The speaker is expressing anger in this audio clip. The reasoning behind this is based on the prosody cues and the semantic content of the speech. The speaker's pitch is high, which is a common feature of anger. The speech speed is also fast, which can indicate frustration or irritation. The energy of the speech is high, which can also be associated with anger. The stress on certain words or phrases can also indicate anger, as the speaker may be emphasizing their point or expressing their frustration. The semantic content of the speech also supports the inference of anger. The speaker is expressing their frustration and annoyance at having to wait in line for an hour. They are also asking if there is any way they can be helped, which can indicate a sense of helplessness or frustration. Overall, the combination of prosody cues and semantic content in the speech supports the inference that the speaker is expressing anger. The high pitch, fast speed, high energy, and stressed words all contribute to this inference. The semantic content of the speech also supports the inference of anger, as the speaker is expressing frustration and annoyance at having to wait in line for an hour. |
| MERaLiON2 (He et al., 2025) | 'The speaker is expressing frustration. Here's why: * Prosody Cues: The speaker's tone is slightly elevated, with a slightly faster pace and a noticeable increase in energy when they say "I just stood in this line for an hour." This indicates agitation and impatience. * Semantic Cues: The phrase "I just stood in line for an hour" conveys a sense of wasted time and frustration. The speaker is clearly unhappy about the situation. * Context: The speaker is likely addressing a customer service representative or someone in authority, as evidenced by the phrase "is there any way that I can..." This suggests a desire for a solution to their problem, further emphasizing their frustration.While the speaker apologizes at the beginning, the overall tone and content of their statement point towards frustration rather than sadness or anger.. |

| DIVA (Held et al., 2024) | Based on the audio clip, I would argue that the speaker is expressing frustration. Here's my reasoning: 1. Prosody cues: The speaker's tone is flat and monotone, with a slightly rising pitch at the end, which suggests a sense of resignation and frustration. The pace is moderate, but the speaker's words are delivered in a slow and deliberate manner, which adds to the sense of frustration. 2. Semantic cues: The speaker's words convey a sense of disappointment and annoyance. The phrase "I stood here for an hour" implies a significant amount of time wasted, and the speaker's tone suggests that they are not pleased about it. 3. Speaker traits: The speaker's age and gender are not explicitly mentioned, but the tone and language used suggest a young adult or an older teenager. The speaker's frustration is likely due to a sense of injustice or unfairness, which is a common emotion among young people. 4. Contextual cues: The speaker is expressing their frustration about waiting in line for an hour, which is a relatable and common experience. The speaker's frustration is likely amplified by the fact that they feel they have wasted a significant amount of time. Overall, the speaker's tone, language, and context all suggest that they are expressing frustration and disappointment about the situation. |
|---|---|
| Megrez-3B-Omni (Infinigence AI, 2024) | The speaker in this audio clip appears to be expressing a sad emotion. Here's a comprehensive reasoning of why this is the case: 1. Prosody Cues: The speaker's intonation is flat and monotonous, which often indicates a lack of enthusiasm or emotional depth. This can be associated with sadness or disappointment. 2. Semantic Cues: The phrase "I'm sorry but I just stood in this line for an hour" suggests that the speaker has experienced a frustrating or disappointing situation. The use of the word "sorry" indicates a sense of regret or remorse, which can be a sign of sadness. 3. Contextual Clues: The fact that the speaker has been standing in a line for an hour suggests that they have been waiting for something or someone, which can be a source of frustration and disappointment. This context, combined with the emotional tone of the speech, supports the interpretation of sadness. 4. Gender and Age Level: While gender and age level are not definitive indicators of emotion, they can sometimes provide additional context. In this case, the speaker's gender and age level are not specified, so they do not significantly influence the interpretation of the emotion. 5. Overall Tone: The overall tone of the speech is subdued and lacks energy, which is often associated with sadness or disappointment. In conclusion, the combination of prosody cues, semantic cues, contextual clues, and the overall tone of the speech suggests that the speaker is expressing a sad emotion. |

| | |
|---|---|
| MiniCPM-O (MiniCPM-o Team, 2024) | The emotion expressed in this audio clip is contempt. Here's a comprehensive analysis: 1. Speaker Traits: The speaker appears to be female, given the voice characteristics and intonation patterns typically associated with women. There are no significant age indicators from the provided information; however, her tone suggests she might be an adult or young adult. 2. Prosody Cues: - Intonation: The pitch of the voice fluctuates slightly but generally remains level without much variation, which can indicate a lack of enthusiasm or engagement. This supports the idea that the speaker does not feel strongly about what they're saying. - Speed: Her speech speed seems normal, neither particularly fast nor slow, suggesting a neutral pace rather than one indicative of strong emotions like excitement (fast) or frustration (slow). - Energy: The energy conveyed through her voice is low-key, lacking high levels of volume or intensity often found in expressions of anger or happiness. 3. Semantic Cues: - Transcription Analysis: "Okay, I'm sorry, but I just stood in this line for an hour" indicates a situation where someone has waited for a long time, likely leading to feelings of annoyance or dissatisfaction. However, the phrase "is there any way that I..." shows politeness and a desire for resolution, hinting at underlying frustration tempered by courtesy. Combining these factors leads us to conclude that the primary emotion being expressed is contempt—specifically, mild disdain or irritation towards having wasted time waiting in line. |
| GLM-4-Voice (Zeng et al., 2024) | In this audio clip, the speaker seems to express exhaustion and possibly frustration. The words "I'm sorry but I just did this for an hour" suggest that they're expressing a sense of relief or weariness, possibly because they've been doing the same task for a long time. The tone of their voice, slow and monotonous, matches the physical exhaustion they describe, further supporting the idea that they're feeling tired and perhaps a bit annoyed |
| OSUM-EChat (Geng et al., 2025) | The emotion expressed in this audio clip is neutral. The speaker is responding in a calm, composed manner. There's no obvious tone of anger, sadness, fear, or surprise. They maintain a polite and even delivery ("I'm sorry but I just stood in line for an hour... Can there be any way that"). The speaker could be regarded as polite and resigned, which generally aligns with a neutral state as they are not conveying strong emotions like joy (happiness) or distress (sadness). |

## Prompt Template (Generate reward model training data )

#### 3. **Caption Completeness**
How thoroughly the explanation covers all relevant metadata and contextual cues (regardless of factual accuracy).
- **5**: Comprehensive coverage of nearly all prosodic and semantic cues.
- **4**: Covers most key elements, though 1–2 important cues may be missed.
- **3**: Mentions several cues but misses critical aspects or remains surface-level.
- **2**: Very limited coverage; ignores most important metadata.
- **1**: Only touches on a single cue or is entirely incomplete.

#### 4. **Fluency & Structure Clarity**
How well-structured, fluent, and easy-to-follow the explanation is (ignoring content accuracy).
- **5**: Very clear, coherent, and well-written; excellent structure and flow.
- **4**: Mostly fluent and understandable; minor grammatical or structural issues.
- **3**: Some awkward or unclear parts; structure may be inconsistent.
- **2**: Hard to follow; poorly organized or expressed.
- **1**: Incoherent or barely readable.

---
### **Requirements:**
- You **must** write the explanation as if you had **only listened** to the audio and **did not** have access to the metadata.
- Like a human listener, use auditory reasoning to deduce the likely emotion, referencing related speaker traits, prosody cues, and semantic(transcript) cues.
- Assume you do **not** know the ground-truth label and suppose you listened to this audio. Like a human listener, infer the most likely emotion through step-by-step reasoning. Let the conclusion arise naturally from your reasoning process. The reasoning process focus should be on **acoustic and prosodic cues**, while semantic content may serve as a secondary reference.**
- **Important:** Your explanation **must reflect** the given score quality exactly:
  - **Factual Alignment** score: `{item['factural_alignment_score']}`
  - **Interpretative Quality** score: `{item['interpretative_quality_score']}`
  - **Caption Completeness** score: `{item['caption_completness_score']}`
  - **Clarity & Coherence** score: `{item['clarity_cohenrences_score']}`

- The reasoning must be **between 50–200 words**.

- **If the target overall quality is low**, you may generate **shorter and less detailed** explanations (closer to 50~100 words).
  **If the overall quality is high**, aim for a **more complete and thorough explanation** (closer to 200 words).
  Adapt the length **proportionally to the overall quality** while staying within the 50–200 words range.

Only return the emotional reasoning content.

Figure 7: Prompt template for generating reward model training data (Part 2).

## Prompt Template (Emotion reasoning evaluation)

You are an expert evaluator for emotion recognition reasoning quality. Your task is to evaluate a model-generated explanation (within <think>...</think>) for its quality in justifying the predicted emotion label (within <answer>...</answer>), using provided metadata and a reference explanation.

Evaluate the explanation across **two dimensions** on a **1–5 scale**:

- **5**: Better than the reference.
- **4**: Good as reference in quality and grounding, allow with few factual mismatch.
- **3**: Weak reasoning, poor structure or many factual errors (Note: if have multiple factual errors should only affect the Factual Alignment score and should not influence other metrics.)
- **2**: Totally irrelevant, serious hallucinated, or no meaningful reasoning
- **1**: Very bad reasoning process.

---
### Context Information

**Metadata of the audio:**
- `groundtruth_emotion`: "{item['major_emotion']}"
- `transcription`: "{item['transcription']}"
- `speaker_gender`: "{item['gender']}"
- `speaker_age`: "{item['age_level']}"
- `pitch_level`: "{item['pitch_level']}"
- `energy_level`: "{item['energy_level']}"
- `speed_level`: "{item['speed_level']}"
- `intonation_contour`: "{item['intonation_contour']}"
- `stressed_word`: "{item['stress']}"

**Reference Explanation:**
{item['gt_reasoning']}

**Model-Generated Explanation (to evaluate):**
{item['pred_reasoning']}

---

### Evaluation Dimensions

1. **Factual Alignment**
   How accurately the explanation uses the provided metadata (e.g., pitch, stress, transcription).
   → Score high if it correctly references facts; penalize hallucinations or omissions.

Figure 8: Prompt template for emotion reasoning evaluation (Part 1).

## Prompt Template (Emotion reasoning evaluation)

You are an expert evaluator for emotion recognition reasoning quality. Your task is to evaluate a model-generated explanation (within <think>...</think>) for its quality in justifying the predicted emotion label (within <answer>...</answer>), using provided metadata and a reference explanation.

...

### Evaluation Dimensions

1. **Factual Alignment**
   How accurately the explanation uses the provided metadata (e.g., pitch, stress, transcription).
    → Score high if it correctly references facts; penalize hallucinations or omissions.

2. **Interpretative Quality**
   Only focus on interpretative. How logically and persuasively the explanation connects acoustic and linguistic features to the emotion.
    → Score high for clear, structured, and convincing reasoning.

3. **Caption Completeness**
   Only focus on overall caption, ignore weather have factual inaccuracies. Whether the explanation covers key metadata points and contextual cues relevant to the emotion.
    → Reward comprehensive coverage; penalize if important cues are ignored.

4. **Clarity & Coherence**
   Only focus on overall coherence. How well-written and easy-to-follow the explanation is.
    → Consider fluency, structure, and conciseness.

---

### Output Format

Only return a JSON object with scores and a concise comment:

```json
{{
 "Factual_Alignment": int,
 "Interpretative_Quality": int,
 "Caption_Completeness": int,
 "Clarity_Coherence": int,
 "comment": "Brief rationale for scores (2-3 sentences)."
}}
```

Figure 9: Prompt template for emotion reasoning evaluation (Part 2).

