# OpenReview forum: "EmotionThinker: Prosody-Aware Reinforcement Learning for Explainable Speech Emotion Reasoning"
_ICLR.cc/2026/Conference — ICLR 2026 Oral_

### Official Review · Reviewer_tjXC · 2025-10-24

**Soundness:** 3
**Presentation:** 3
**Contribution:** 3
**Rating:** 8
**Confidence:** 2

**Summary:**

This paper introduces **EmotionThinker**, a novel model for speech emotion reasoning that aims to reframe Speech Emotion Recognition (SER) from a simple classification task into a deep, explainable reasoning problem. The core contribution is the design of a **Prosody-Aware Reinforcement Learning (RL) framework**. This framework guides a Large Language Model (LLM) to generate coherent, feature-grounded text explanations (i.e., reasoning paths), thereby bridging raw acoustic signals, the textual reasoning process, and the final emotional label. This innovative approach addresses the critical lack of interpretability in existing SER systems and SpeechLLMs.

**Strengths:**

- **Originality (Originality):** Very High. The combination of SER, LLM, and RL specifically tailored for generating prosody-grounded explanations is highly novel within the speech community.
- **Quality (Quality):** High. The approach moves beyond simple performance metrics by incorporating explanation quality into the optimization objective, indicating a rigorous research focus on a complex problem.
- **Clarity (Clarity):** Good. The core idea is presented clearly, and the reasoning process (via case studies) is easily digestible.
- **Significance (Significance):** Substantial. This work significantly pushes the boundaries of transparency and trust in SpeechLLMs for emotional tasks, which is an important step for the future of multimodal AI.

**Weaknesses:**

1. **Technical Granularity of RL and Prosody Integration (Critical):**
	- The paper emphasizes **"Prosody-Awareness,"** but the precise and **explicit mechanism** by which the RL framework guides the LLM to attend to the *most critical* prosodic features (e.g., sudden pitch shifts over average pitch) needs more profound elaboration.
	- The **RL Reward Function** is paramount. A detailed ablation study is essential to show how the different components of the reward (classification accuracy vs. fluency vs. factual grounding to acoustic features) are balanced and how this balance impacts the quality and faithfulness of the final explanation. The current description suggests this crucial balance may be underspecified.
2. **Data and Generalization Concerns:**
	- Training LLMs via RL for generation tasks often relies heavily on high-quality **human-annotated reasoning path data** for initial Supervised Fine-Tuning (SFT) or as part of the reward signal. The paper must provide a candid discussion on the cost and scarcity of this data and how the model manages to generalize its reasoning to novel or atypical speech examples outside the training distribution.
	- Generalization to different languages or accents, crucial for a model involving LLM-style reasoning, is also a concern that needs addressing.
3. **Efficiency and Deployment Feasibility (Practicality):**
	- The combination of LLM and RL training typically incurs a substantial computational overhead. The paper is currently lacking in a detailed analysis of the **training efficiency, required computational resources (GPU-hours)**, and most importantly, the **inference latency** compared to existing, lightweight SER systems. This is vital for assessing the model's practical viability for real-world deployment.

**Questions:**

1. RL Reward Faithfulness and Ablation (Critical)

	Provide an **Ablation Study** on the RL reward components. How do you ensure the explanations are **truly faithful to prosodic facts** and not just syntactically fluent fabrications?

2. Technical Mechanism of Prosody Grounding

	Clarify the **explicit mechanism** that links a generated text token (e.g., "high pitch") to the **specific, salient acoustic feature** in the speech input.

3. Practicality and Efficiency Analysis

	Provide detailed **Inference Latency** and **Training Cost (GPU-hours)** analysis. Is this model practically deployable compared to lightweight SER baselines?

4. Generalization

	Show **Out-of-Distribution (OOD)** results to prove that the learned reasoning generalizes beyond the training corpus.

---

> ### Author Response · Authors · 2025-11-21
> **Author Response to Reviewer tjXC (1/2)**
>
> Dear Reviewer tjXC,
>
> Thank you for the valuable comments and encouraging feedback! We appreciate your recognition of our contributions. Since several points in the Weakness and Question sections overlap, we merge them into the following five core issues to avoid redundancy and address them clearly.
>
> **W1&Q1: Technical Granularity of RL and Prosody Integration (e.g., How do you ensure the explanations are truly faithful to prosodic facts and not just syntactically fluent fabrications?)**
>
> To address whether the explanations are truly grounded in prosodic evidence rather than syntactically fluent fabrications, we provide a deeper ablation study on the reward components. In Table R5, when we remove prosody-related dimensions (Factual Alignment and Caption Completeness) and keep only linguistic-surface dimensions (Interpretative Quality and Fluency/Structure), both the overal reasoning score and the emotion accuracy drop. This indicates that simply rewarding syntactic fluency cannot maintain faithful prosody alignment.
>
> We also provide ablations over the full RL framework each components: (i) rule-based accuracy reward only, without reasoning supervision (Table 4-V2 in paper), (ii) untrained reward model for prosody supervision (Table 4-V3), (iii) removing the progressive schedule (Table 4-V5), and (iv) removing the reasoning trust-awareness weight (Table 4-V6). Across all variants, each component contributes meaningfully and removing any of them degrades both reasoning quality and emotion accuracy. Table 7 in paper further examines how different reward components are balanced and how this balance affects emotion accuracy. Taken together, these results show that GRPO-PTR enforces prosody-grounded reasoning rather than surface-level fluent text generation.
>
> Table R5: Ablation on prosody-grounded reward components.
> | Variation | Emotion Acc. | Emotion Reaoning (Avg) | FA. | CC. | IQ. | FS. |
> |----------|----------|----------|----------|----------|----------|----------|
> | Baseline | 68.89%  | 3.98 | 3.54  | 3.96 |4.01 | 4.42 |
> | prosody-aware reward signals  | 66.43%  | 3.73 | 3.04 | 3.12 | 4.26 | 4.48 |
>
> **W2: Data and Generalization Concerns (Including providing discussion on the cost and scarcity of this data, generalization to different languages or accents)**
>
> All reasoning paths in EmotionCoT-35K are generated via a fully automatic labeling pipeline, without human annotation cost. We adopt this design to achieve scalable data construction, reduced annotation cost, and stable label quality. Prosodic attributes (e.g., pitch, energy, intonation) are obtained with deterministic signal-processing methods, eliminating ambiguity in these labels. For speaker-trait annotations (e.g., gender and age group), most emotion datasets already provide these metadata; when not available, we supplement them using widely adopted pretrained classifiers with reported accuracies above 90% on public benchmarks [1].
>
> Regarding generalization, prosodic cues such as pitch movement, stress, and intensity variation are language-agnostic acoustic phenomena, and our RL framework does not rely on any language-specific assumptions. Therefore, both our data construction method and the GRPO-PTR training paradigm are directly applicable to other languages and accents. We will include a clearer discussion of these points in the revised version.
>
> **W3&Q3: Efficiency and Deployment Feasibility (Provide detailed inference latency and training cost analysis. Is this model practically deployable compared to lightweight SER baselines?)**
>
> For training efficiency, The SFT stage was trained on 4×A100 GPUs for 6 hours, and the RL stage required 8×A100 GPUs for one day. We will provide the training and inference details in the revised version.
>
> For inference latency, EmotionThinker uses a 7B LLM backbone to generate both the reasoning trace and the final label. This makes the runtime dependent on the generated response length due to autoregressive next-token decoding. On an A100 GPU, the end-to-end inference time is typically 1~3 seconds per sample (including audio encoding and ~200 generated tokens), which is expected for 7B-class LLMs. Consequently, EmotionThinker is not directly comparable to lightweight classification-based SER systems that require only a single forward pass and do not generate textual explanations.

---

> ### Author Response · Authors · 2025-11-21
> **Author Response to Reviewer tjXC (2/2)**
>
> **Q2: Clarify the explicit mechanism that links a generated text token (e.g., "high pitch") to the specific salient acoustic feature in the speech input.**
>
> Our framework enforces prosody grounding through three aligned stages:
>
> **(1) Data construction (section 3.1):** for each audio, it’s salient acoustic features will be labelled and uesed as contextual cues when generating reasoning races, so the supervision signals are inherently aligned with the original audio.
>
> **(2) SFT stage (section 3.2):** we explicitly train the backbone (EmotionThinker-Base) to discriminate fine-grained prosodic patterns (e.g., intonation trends, speed variations), which equips the model with strong prosody perception before RL. This prosody-awareness is the foundation that enables the model to later generate interpretable and acoustically grounded reasoning.
>
> **(3) RL stage (section 3.3):** during GRPO-PTR training, the reward model explicitly evaluates whether each generated reasoning candidate correctly reflects the actual salient acoustic events detected from the input audio. Candidates that accurately describe prosodic facts receive high factual-alignment and completeness rewards, while mismatched or hallucinated prosody descriptions are penalized. This systematically encourages the model to generate text tokens that are aligned with true acoustic evidence, rather than surface-level fluent fabrications.
>
> **Q4: Show Out-of-Distribution (OOD) results to prove that the learned reasoning generalizes beyond the training corpus.**
>
> As shown in our main Table 2,  RADESS and SAVEE set are Out-of-Distribution (OOD) results, only IEMOCAP and MELD training set are used during training. On these OOD corpora, as we can see that in emotion recognition, we achieve best performance on SAVEE and second-best performance on RADESS among 16 different SpeechLLMs. And in emotion reasoning, we have higher scores on different dimensions.
>
> ---
> Reference:
>
> [1] Felix Burkhardt, et al. "Speech-based Age and Gender Prediction with Transformers." arXiv preprint arXiv:2408.13608 (2024).

---

### Official Review · Reviewer_dYpE · 2025-10-28

**Soundness:** 3
**Presentation:** 3
**Contribution:** 3
**Rating:** 6
**Confidence:** 4

**Summary:**

This paper proposes EmotionThinker, a novel framework for explainable speech emotion recognition (SER) that leverages CoT reasoning and reinforcement learning (RL) to move beyond standard categorical classification. The authors introduce EmotionCoT-35K, a large dataset with CoT annotations and fine-grained prosodic and semantic factors tailored for emotion reasoning. They further proposed an RL-based optimization framework (GRPO-PTR) that incorporates a progressive, trust-aware reasoning reward, balancing outcome accuracy and reasoning quality. Extensive experiments over four benchmarks and ablation studies demonstrate that EmotionThinker achieves superior emotion recognition accuracy and richer, more interpretable explanations compared to a wide range of baselines.

**Strengths:**

1. The reformulation of SER as a deep reasoning task—rather than mere label prediction—is timely and promising for advancing interpretability in multimodal LLMs.
2. The proposed dataset, EmotionCoT-35K, fills a significant gap with CoT-style, prosody-aware emotion reasoning data, with a scalable, largely automated annotation pipeline. This may have value for the broader community.
3. The proposed reinforcement learning scheme employs progressive reward scheduling and a trustworthiness weight to dynamically balance outcome and reasoning reward signals. This helps mitigate reward hacking and stabilizes training and may be meaningful for the LLM RL community.

**Weaknesses:**

1. The data construction pipeline heavily relies on LLMs, and the reasoning trace data is constructed with GPT4o without the actual speech input. This may lead to unexpected failure and bias in the dataset. It would also be beneficial to input the speech and conduct a human review of the data quality.
2. The proposed reward model plays a critical role in the RL process. However, there is little discussion or quantitative validation of its calibration. The distributions of GPT-annotated versus human-annotated reward scores are not directly compared.
3. The description of the RL part is not very easy to follow. It would be better to improve the logit flow in this part.

**Questions:**

1. Can the authors provide more analysis comparing the similarities and differences between GPT-4o-based and human-based scoring for CoT data and reasoning reward trace quality? Are there specific failure modes or biases in the model-synthesized data?
2. Are there statistics on annotation accuracy for the automated annotations (prosody, stress, speaker traits) used in EmotionCoT-35K? Do certain emotion categories or speaker groups have systematically noisier annotations or explanations?
3. Typos: Line 270, Appendix without reference.

---

> ### Author Response · Authors · 2025-11-21
> **Author Response to Reviewer dYpE**
>
> Thank you for the valuable comments and encouraging feedback! We appreciate your recognition of our contributions. Please find our response to the specific points below.
>
> **W1: The reasoning trace data is constructed with GPT4o without the actual speech input**
>
> Thank you for raising this point. During data construction, we did use the speech input and test several state-of-the-art SpeechLLMs (e.g., GPT-4o-audio, Qwen2.5-Omni, Gemini-2.5-Pro), but they still struggled to accurately interpret fine-grained prosodic cues. Based on this observation, we adopted a different strategy: we first extract accurate acoustic features using traditional speech-processing tools and then provide these cues as structured context for GPT-4o to generate prosody-aware reasoning traces. We also conducted human reviews on sampled data to assess the alignment between the generated reasoning and the actual audio. In practice, we found that this hybrid pipeline leads to higher reasoning quality and more faithful prosody-grounded explanations.
>
> **W2&Q1: More analysis about the alignment between human-based and model-based scores.**
>
> To validate the alignment between GPT-4o-based scores and human judgments, we evaluated 100 model-generated reasoning candidates by comparing the reward model’s scores with the averaged ratings from three human annotators. As shown in Table R4, the reward model shows generally good alignment with human ratings across all four criteria. We also observe that both human and model scores follow similar ranking trends across samples, indicating that the reward model captures relative quality reliably.
>
> Table R4: Correlation analysis between reward-model scores and human ratings across 100 sampled reasoning candidates.
> | Criterion | Reward Model Mean | Human Mean | Correlation |
> |----------|----------|----------|----------|
> | Factual Alignment (FA) | 3.10  | 3.5  | 0.87 |
> | Interpretative Quality (IQ)  | 3.95  | 3.89  | 0.82 |
> | Caption Completeness (CC)  | 3.03  | 3.22  | 0.79 |
> | Fluency & Structure (FS)  | 4.30  | 4.35  | 0.91|
>
> **W3: The description of the RL part is not very easy to follow.**
>
> Thanks for the feedback. We will revise the RL section by enhancing its structure, adding more details, and providing clearer explanations to improve readability.
>
> **Q2: Are there statistics on annotation accuracy for the automated annotation? Do certain emotion categories or speaker groups have systematically noisier annotations or explanations?**
>
> The prosody-related annotations such as pitch, energy, and speaking speed are computed using deterministic speech-processing tools with given thresholds. Therefore, these features are extracted directly from the raw audio and do not involve classification uncertainty. For speaker-trait annotations (e.g., gender and age group), most emotion datasets already provide these metadata; when not available, we supplement them using widely adopted pretrained classifiers with reported accuracies above 90% on public benchmarks [1]. We will include these details in the revised version. Prosody-based features remain stable across categories because they are extracted deterministically. We also sample-checked a subset of the data and did not observe systematic noise patterns or category-specific error trends.
>
> **Q3: Typos- Line 270, Appendix without reference.**
>
> Thank you for pointing this out. We will fix the typo at Line 270 and add the missing appendix reference in the revised version.
>
> ---
> Reference:
>
> [1] Felix Burkhardt, et al. "Speech-based Age and Gender Prediction with Transformers." arXiv preprint arXiv:2408.13608 (2024).

---

### Official Review · Reviewer_uQyK · 2025-10-31

**Soundness:** 3
**Presentation:** 4
**Contribution:** 3
**Rating:** 6
**Confidence:** 5

**Summary:**

First, I would like to summarize the contributions of the work by reading the abstract.

First, a speech emotional dataset was constructed. Second, current speech LLMs have weak prosody perception. This work tries to address this issue by developing a prosody-enhanced foundation model. Third, a new type of reinforcement learning protocol is proposed, which progressively introduces a reasoning award by dynamically adjusting it with trustworthy weights, reflecting the alignment between reasoning and outcome.

**Strengths:**

**1.** The motivation of the work is clearly stated and explained.

**2.** A first RL-based emotion recognition that has the ability not only for accurate classification, but detailed reasoning rationales and informative captions for the audio.

**3.** Each stage of the proposed framework is clearly defined.

**4.** The evaluation and abolition are comprehensive.

**Weaknesses:**

**1.** For the accuracy of emotion recognition, I would also like to know the performance on each individual discrete emotion. That way, we can have a more concrete and detailed understanding of the framework's capabilities and limitations.

**2.** To construct the reasoning responses, is there a specific reason that only GPT 4.0 is used?

**Questions:**

**Other comments:**

For section 3.1, the authors discussed the open-sourced datasets they used to construct the EmotionCot-35k.  I think authors could have a brief discussion on other related, multimodal datasets in the paper as well, whether to use them for constructing the new dataset or not. Such as the following:

MSP-PODCAST (the most recent, final version): Busso, Carlos, et al. "The msp-podcast corpus." arXiv preprint arXiv:2509.09791 (2025).

MERSA: Zhang, Enshi, Rafael Trujillo, and Christian Poellabauer. "The MERSA dataset and a transformer-based approach for speech emotion recognition." Proceedings of the 62nd Annual Meeting of the Association for Computational Linguistics (Volume 1: Long Papers). 2024.

CMU-MOSEI: Zadeh, AmirAli Bagher, et al. "Multimodal language analysis in the wild: Cmu-mosei dataset and interpretable dynamic fusion graph." Proceedings of the 56th Annual Meeting of the Association for Computational Linguistics (Volume 1: Long Papers). 2018.

DECAF: Abadi, Mojtaba Khomami, et al. "DECAF: MEG-based multimodal database for decoding affective physiological responses." IEEE Transactions on Affective Computing 6.3 (2015): 209-222.

---

> ### Author Response · Authors · 2025-11-21
> **Author Response to Reviewer uQyK**
>
> Dear Reviewer uQyK,
>
> Thank you for the valuable comments and encouraging feedback! We appreciate your recognition of our contributions. Please find our response to the specific points below.
>
> **W1: The performance on each individual discrete emotion.**
>
> Thank you for the suggestion. We report the per-emotion accuracy of EmotionThinker aggregated across the four evaluation benchmarks in Table R2. Performance is strongest on angry, happy, neutral, and sad, while surprise and contempt/disgust show lower accuracy, which is consistent with the variability and sparsity of these categories across benchmarks. We will include this per-emotion breakdown in the revised version.
>
> Table R2: Per-class emotion recognition accuracy (%) across all four benchmarks.
>
> | Model | Angry | Happy | Neutal | Sad | Surprise | Comtempt or Disgust | Fear |
> |----------|----------|----------|----------|----------|----------|----------|----------|
> |EmotionThinker| 65.85% | 74.4% | 79.31% | 68.64% | 45.10% | 41.06% | 56.37% |
>
>
> **W2:  To construct the reasoning responses, is there a specific reason that only GPT 4.0 is used?**
>
> We also experimented with several alternative models (e.g., GPT-3.5-turbo, DeepSeek) when constructing the reasoning responses. In practice, we found that the GPT-4o API consistently produced higher-quality, more causally coherent emotional reasoning for our task. For this reason, we adopted GPT-4o as the primary generator in the current version.
>
> **Q1: Have a brief discussion on other related, multimodal datasets in the paper.**
>
> Thank you for this helpful suggestion. These datasets are also valuable resources. We will add more discussion in Section 3.1. Below we summarize key annotations in comparison with our dataset (Table R3). As shown, existing multimodal datasets provide useful emotion labels but generally lack fine-grained prosodic descriptors or reasoning-style annotations required by our CoT-based pipeline.
>
> Table R3: Comparison with other multimodal emotion datasets.
> | Dataset         | Reasoning | Age | Gender | Emotion | Pitch | Speed | Energy | Style | Contour | Stress |
> |-----------------|-----------|-----|--------|---------|-------|--------|--------|--------|---------|---------|
> | MSP-PODCAST[1]     |     ✗     |  ✓  |   ✓    |    ✓    |   ✗   |   ✗    |   ✗    |   ✓    |    ✗    |    ✗    |
> | MERSA[2]          |     ✗     |  ✓ |   ✓    |    ✓    |   ✗   |   ✗    |   ✗    |   ✗    |    ✗    |    ✗    |
> | CMU-MOSEI[3]       |     ✗     |  ✗  |   ✓    |    ✓    |   ✗   |   ✗    |   ✗    |   ✓    |    ✗    |    ✗    |
> | DECAF[4]           |     ✗     |  ✓  |   ✓    |    ✓    |   ✗   |   ✗    |   ✗    |   ✓    |    ✗    |    ✗    |
> | EmotionCoT-35K           |     ✓     |  ✓  |   ✓    |    ✓    |   ✓   |   ✓    |   ✓    |   ✓    |    ✓    |    ✓   |
>
> ---
> **References:**
>
> [1] MSP-PODCAST (the most recent, final version): Busso, Carlos, et al. "The msp-podcast corpus." arXiv preprint arXiv:2509.09791 (2025).
>
> [2] MERSA: Zhang, Enshi, Rafael Trujillo, and Christian Poellabauer. "The MERSA dataset and a transformer-based approach for speech emotion recognition." Proceedings of the 62nd Annual Meeting of the Association for Computational Linguistics (Volume 1: Long Papers). 2024.
>
> [3] CMU-MOSEI: Zadeh, AmirAli Bagher, et al. "Multimodal language analysis in the wild: Cmu-mosei dataset and interpretable dynamic fusion graph." Proceedings of the 56th Annual Meeting of the Association for Computational Linguistics (Volume 1: Long Papers). 2018.
>
> [4] DECAF: Abadi, Mojtaba Khomami, et al. "DECAF: MEG-based multimodal database for decoding affective physiological responses." IEEE Transactions on Affective Computing 6.3 (2015): 209-222.

---

> > ### Comment · Reviewer_uQyK · 2025-11-24
> > **Response to author's rebuttal - from Reviewer uQyK**
> >
> > Thank you for your responses. Based on the results presented in Table R2, it appears that the high accuracy for certain emotional labels may be due to the abundance of data points associated with those labels. It is somewhat surprising to see that the emotion 'neutral' achieved high accuracy, as this category is typically classified alongside more ambiguous emotions like happiness.
> >
> > If you have experimented with other models, please include those in your revised version as well.
> >
> > For the comparison with other multimodal SER datasets, you can incorporate them into your current Table 1 or create a separate table, depending on which option provides clearer presentation.
> >
> > Overall, I believe these responses will further enhance the paper's quality. I will update my score to 8.

---

> > > ### Author Response · Authors · 2025-11-26
> > > **Author Response to Reviewer uQyK**
> > >
> > > Thanks for your kind reply and recognition! In the revised version, we will (i) explicitly discuss how data imbalance contributes to performance differences across emotions, and (ii) add results from additional baseline models to provide a more comprehensive performance comparison. Regarding multimodal SER datasets, we will integrate the comparison into Table 1 for better comparison. Thank you again for the detailed feedback and positive evaluation. Your comments greatly help us improve the clarity and completeness of the paper.

---

### Official Review · Reviewer_Ez3y · 2025-11-01

**Soundness:** 3
**Presentation:** 2
**Contribution:** 3
**Rating:** 6
**Confidence:** 4

**Summary:**

The paper explores an interesting problem by extending emotion modeling from classification to reasoning with promising results. However, the methodology and training details are unclear, reproducibility is lacking due to missing code release, and definitions of prosody and emotional cues need stronger justification and clarity.

**Strengths:**

The motivation is clear, and the research problem is interesting, as it extends beyond improving emotion classification toward developing deeper reasoning capabilities.

The proposed model demonstrates strong performance in both emotion recognition and emotion reasoning, providing valuable insights for advancing SpeechLLMs toward more effective emotion reasoning capabilities.

**Weaknesses:**

It is unclear how your model is trained and how it builds upon Qwen2.5-Omni-3B. Please clarify the training process and provide clear explanations for all symbols and notations in your equations, as they are currently difficult to interpret.

The methodology section, particularly Section 3.3, lacks clarity. Please provide a clear description of the overall training pipeline and explain the motivation behind each step. The writing in Section 3.3.1 should be further improved for better structure and readability. Additionally, clarify the purpose of the forward reward and outcome accuracy reward, why both are needed, how they relate to the components shown in Figure 3, and what their specific inputs and outputs are.

Will you release your code and dataset? The reproducibility checklist is missing, and without a clear commitment to open-sourcing these resources, the paper’s reproducibility and credibility are severely limited. I may not be able to recommend acceptance unless this issue is properly addressed.

**Questions:**

How do you handle emotional cues that arise from linguistic content? For example, if the text conveys a happy emotion but the corresponding speech expresses sadness, which modality is prioritized in your final emotion prediction? Does emotion inferred from text affect the overall performance of your model?

Please provide both theoretical justification and experimental evidence to support your claim that “prosodic signals are core carriers of emotional intent.” How do you account for the role of textual content and nonvocal components (e.g., crying, laughter)? If you argue that prosody is the most dominant factor, please include empirical evidence demonstrating that prosody contributes more significantly to emotion perception than textual and nonvocal cues.

Please clarify how you define prosody. Are speaker traits such as gender and age also included under this term? Appropriate literature references should be provided to accurately define both “prosody” and “speaker traits,” as some of the current definitions appear to be inaccurate.

---

> ### Author Response · Authors · 2025-11-21
> **Author Response to Reviewer Ez3y (1/3)**
>
> Dear Rewier Ez3y,
>
> Thank you for the valuable comments and encouraging feedback! We appreciate your recognition of our contributions. Please find our response to the specific points below.
>
> **W1: It is unclear how your model is trained and how it builds upon Qwen2.5-Omni-3B.**
>
> We apologize for the confusion. We want to clarify that EmotionThinker does not build upon Qwen2.5-Omni-3B, and our system contains two components based on the Qwen2.5-Omni family, and we will clarify this more explicitly in revised version.
>
> - Main model (EmotionThinker).
>
> EmotionThinker is built upon Qwen2.5-Omni-7B. We first perform prosody-centric SFT on prosody-related tasks, followed by 5k EmotionCoT cold-start training to obtain EmotionThinker-Base. This jointly updates the speech encoder, adapter, and LLM. We then apply GRPO-PTR reinforcement learning on EmotionCoT-35K, where EmotionThinker-Base is optimized as the policy model.
>
> - Reasoning reward model (based on Qwen2.5-Omni-3B).
>
> Separately, we train a reward model built upon Qwen2.5-Omni-3B. It is fine-tuned on 101,400 (q, r, g) tuples, where *q* is the emotional prompt, *r* is a model-generated reasoning trace, and *g* is a 4-dimensional vector of criterion-specific scores (factual alignment, interpretative quality, caption completeness, fluency/structure). These tuples are constructed by extending EmotionCoT-35K with GPT-4o-generated reasoning variants of controlled quality. During GRPO-PTR training, this 3B reward model is used only for scoring and remains frozen.
>
> - Notation clarification.
>
> We will explicitly define all symbols in the revised version:
>
> $R^f$ (format reward), $R^o$ (outcome accuracy reward), $R^t$ (reasoning reward assigned by the 3B reward model), $\tau$ (trustworthiness weight), and the final reward:
>
> $$
> R_i = \alpha_f R^f + \alpha_o R^o + \alpha_t \tau R^t.
> $$
>
> We will also clarify all symbols used in the computation of $\tau$:
>
> - $G_{\text{correct}}$: the set of model-generated responses whose *outcome label* is correct.
> - $G_{\text{wrong}}$: the set of model-generated responses whose *outcome label* is incorrect.
> - $R_t^{(c)}$: the average reasoning reward $R^t$ assigned to all responses in $G_{\text{correct}}$.
> - $R_t^{(w)}$: the average reasoning reward $R^t$ assigned to all responses in $G_{\text{wrong}}$.
>
> These quantities are used to compute the trustworthiness weight $\tau$, which downweights cases where incorrect responses are mistakenly assigned high reasoning rewards.
>
>
> **W2. The methodology section, particularly Section 3.3, lacks clarity. Please provide a clear description of the overall training pipeline and explain the motivation behind each step.**
>
> Thank you for the feedback. We will improve Section 3.3, especially Section 3.3.1, by enhancing its structure, adding more details, and providing clearer explanations to improve readability. Below we address each aspect of your comment.
>
> - Training pipeline and purpose.
>
> EmotionThinker is trained in two stages. (1) We first conduct prosody-centric SFT on Qwen2.5-Omni-7B backbone, followed by 5k EmotionCoT cold-start samples. This stage strengthens prosody perception and teaches the model the basic reasoning format, producing EmotionThinker-Base. (2) We then apply GRPO-PTR reinforcement learning on EmotionCoT-35K, where EmotionThinker-Base is the trainable policy model, and the reward model provides scores for each generated reasoning trace. In this stage, both reasoning quality and emotion accuracy are jointly optimized to produce the final EmotionThinker model.
>
> - Purpose of format reward and outcome accuracy reward.
>
> The format reward ensures the model produces valid structured outputs so that reasoning scores can be computed reliably. The outcome accuracy reward directly aligns the final emotion prediction with ground-truth labels. Both are needed: the former stabilizes generation, while the latter prevents the model from producing well-formed but incorrect answers.
>
> - Relation to Figure 3 and I/O definition.
>
> In Figure 3, the format reward (R_f in Fig.3) takes the generated text as input and outputs a binary scalar: it returns 1 when the output matches the required structure `<think>(reasoning)</think><answer>happy</answer>` and 0 otherwise.; the outcome reward (R_o in Fig.3) evaluates the `<answer>` segment by comparing the predicted emotion label with the ground truth.
>
> **W3: Will you release your code and dataset?**
>
> Yes, we will release our model, code, and data after acceptance.

---

> ### Author Response · Authors · 2025-11-21
> **Author Response to Reviewer Ez3y (2/3)**
>
> **Q1: How do you handle emotional cues that arise from linguistic content?**
>
> Good question. When textual and acoustic cues conflict, we follow the convention in speech-emotion datasets, where ground-truth labels are determined by the acoustic emotion rather than the literal transcript. As seen in phenomena like sarcasm or ironic speech, the true emotion is often conveyed by tone while the literal words express the opposite meaning. Thus, in our design, speech-based cues are prioritized, and the RL rewards explicitly encourage the model to ground its reasoning in acoustic evidence. Text still provides useful context, but it does not dominate the final prediction when the two modalities disagree.
>
> **Q2: Provide both theoretical justification and experimental evidence to support your claim that “prosodic signals are core carriers of emotional intent.”**
>
> We appreciate the reviewer’s comment. We agree that emotion can be inferred from multiple dimensions (e.g., semantics, nonvocal cues). Our intention was not to claim that prosody is the *most dominant* cue, but rather that prosodic patterns constitute a *key carrier* of emotional intent. In practice, the classical speech-emotion datasets used in our study (e.g., IEMOCAP, MELD, Expresso), textual emotion is typically neutral and nonvocal events are largely absent. As a result, prosody becomes the most interpretable and informative cue in our setting.
>
> 1.Experimental evidence.
>
> Models trained with prosody-perception enhancement during SFT consistently achieve a higher upper bound after RL compared with models without prosody enhancement under identical settings (Table R1). Ablations in Table 4 (in the main paper) further show that, relative to standard GRPO focusing only on emotion prediction (V2), the GRPO-PTR variants (V3–V6) achieve consistently better performance once the reasoning reward explicitly evaluates prosodic cues. These results indicate that prosodic information is an important driver of emotion understanding.
>
> 2.Theoretical justification.
>
> Prior work [1-8] in psycholinguistics and affective science suggests that prosodic modulations can encode emotional dimensions such as valence, arousal, and certain discrete categories. Existing studies also report that listeners are often able to infer emotional intent even when lexical content is neutral or semantically limited. These findings motivate our decision to explicitly model pitch, energy, and temporal dynamics as part of the emotional signal.
>
> 3.Intuitive explanation.
>
> Human listeners naturally rely on prosodic cues—pitch variation, intensity, speaking rate, and intonation—to judge affect. For example, anger often presents with high energy and fast tempo, while sadness tends to exhibit flatter pitch and lower intensity. Similarly, our work is designed to encourage the model to mimic this human-like process of emotional interpretation.
>
> We will revise the paper to use more precise phrasing (“key carriers” instead of “core carriers”) and add the expanded analyses to avoid overstating the role of prosody.
>
> Table R1: Emotion recognition accuracy (%) of EmotionThinker with (`w/ prosody-sft`) and withought (`w/o prosody-sft`) prosody-perception enhancement during SFT.
>
> | Version | IEMOCAP | MELD | RADESS | SAVEE | Avg |
> |----------|----------|----------|----------|----------|----------|
> | w/o prosody-sft | 72.10  | 51.86  | 65.57 | 68.46 | 62.55 |
> | w/ prosody-sft  | 77.68  | 59.71  | 71.56 | 73.96 | 68.89 |
>
> **Q3: Clarify the definination of prosody and speaker traits.**
>
> Thank you for pointing this out. In linguistics [9], prosody refers to suprasegmental properties such as pitch, intensity, duration, rhythm, and intonation, and does not include speaker traits. Gender, age, and timbre are instead categorized as speaker traits, following standard definitions in phonetics [9]. We will revise the paper to include the appropriate references and clearly separate these concepts.

---

> > ### Author Response · Authors · 2025-11-21
> > **Author Response to Reviewer Ez3y (3/3)**
> >
> > References:
> >
> > [1] Paulmann, Silke, and Sonja A. Kotz. “Valence, arousal, and task effects in emotional prosody processing.” *Frontiers in Psychology* 4 (2013): 222.
> >
> > [2] Larrouy-Maestri, Pauline, et al. “The sound of emotional prosody: nearly three decades of research.” *Psychonomic Bulletin & Review* (2024).
> >
> > [3] Lausen, Anja, and Kurt Hammerschmidt. “Emotion recognition and confidence ratings predicted by vocal stimulus type and prosodic parameters.” *Humanities and Social Sciences Communications* 7.1 (2020): 1–11.
> >
> > [4] Nygaard, Lynne C., and D. Robert Ladd. “The semantics of prosody: Acoustic and perceptual aspects.” *Cognitive Science* 33.1 (2009): 51–86.
> >
> > [5] Kadali, Devi Bhavani, and Vinay Kumar Mittal. “Studies on paralinguistic sounds, emotional speech and expressive voices.” *Proceedings of the Workshop on Speech, Music and Mind (SMM)*, 2020.
> >
> > [6] Cao, Houwei, et al. “Prosodic cues for emotion: Analysis with discrete characterization of intonations in the ToBI framework.”
> >
> > [7] Liu, Tong, and Xiaochen Yuan. “Paralinguistic and spectral feature extraction for speech emotion classification using machine learning techniques.” *EURASIP Journal on Audio, Speech, and Music Processing* 2023 (2023): 23.
> >
> > [8] Pepino, Leonardo, Pablo Riera, Luciana Ferrer, and Agustin Gravano. “Fusion approaches for emotion recognition from speech using acoustic and text-based features.” arXiv preprint arXiv:2403.18635 (2024).
> >
> > [9] Crystal, David. *A Dictionary of Linguistics & Phonetics*, 6th ed. Oxford: Blackwell Publishing, 2008.

---

### Author Response · Authors · 2025-12-01
**Summary of the Work and Rebuttal for the Area Chair**

Dear Area Chair,

Thank you so much for taking the additional time to review our paper. We are writing to provide a brief summary of our work and the rebuttal for your convenience.

**Work Summary:** In this work, we propose EmotionThinker, a novel RL framework that take the first step expands speech emotion recognition (SER) from traditional simple classification to explainable emotion reasoning. To answer the core question - *Can SpeechLLMs reason like human about “why” they make emotional judgements?* -  we first construct a fine-grained prosody labeling dataset, EmotionCoT-35K. Second, we develop the prosody-enhanced foundation model EmotionThinker-Base that fills the current gap of weak prosody perception. Finally, we propose GRPO-PTR (Group-Relative-Policy Optimization with Progressive Trust-aware Reasoning Reward), which supervises emotion reasoning process on multiple dimensions while mitigating potential reward hacking caused by misalignment between reasoning quality and outcomes. Our model EmotionThinker offers the following advantages: (1) higher emotion recognition accuracy; (2) deep reasoning ability to integrate emotion-related cues for justification; (3) fine-gained audio caption covering speaker traits, prosodic cues and semantic information.

**Recognition from all reviewers:** We are pleased to see consistently positive feedback (original score: 8/6/6/6 → rebuttal period before openreview issue: 8/8/6/6). We highlight comments from reviewers:

- **Community contribution:** “value for the broader community”, “first RL-based emotion recognition...”, “is timely and promising”, “Originality: Very High”, “research problem is interesting”.
- **Method:** “highly novel”, “helps mitigate reward hacking and stabilizes training”, “more effective”, “has the ability not only for accurate classification, but detailed reasoning rationales and informative captions for the audio”, “fills a significant gap”.
- **Experiment:** ”strong performance in both emotion recognition and emotion reasoning”, “valuable insights“, “evaluation and abalition are comprehensive”.

**Rebuttal Updates:** During the rebuttal period, we provided detailed responses and clarified our work, mainly focusing on (1) improving the clarity of several writing components as requested by the reviewer (reviewer Ez3y) (2) providing supplementary details including per-emotion performance (reviewer uQyK) and ablation results (reviewers dYpE & tjXC) (3) additional information including training and inference efficiency (reviewer tjXC) and data curation details (reviewer dYpE). We sincerely thank the reviewers for the constructive feedback, which greatly helped us improve the paper.

We hope EmotionThinker contributes to advancing the field toward more interpretable, human-aligned speech understanding systems. Thanks again for your time and careful consideration!

Best regards,

The Authors of Submission 2425

---

### Meta-Review · Area_Chair_pdZv · 2026-01-07

**Summary:**

This paper proposes reframing speech emotion recognition as explainable emotion reasoning, rather than categorical classification. It presents a prosody-enhanced foundation model trained on a newly constructed dataset (EmotionCoT-35K) with chain-of-thought annotations and detailed captions. To address weak prosody perception in existing speech LLMs, the authors design a prosody-aware reinforcement learning framework (e.g., GRPO-PTR) that progressively incorporates a trustworthiness weight in the reward computation, with the aim of balancing outcome accuracy with reasoning quality. Extensive experiments and ablations across multiple benchmarks show improved performance and more interpretable explanations.

**Reviewer Concerns:**

Reviewers raised a set of weaknesses, which the authors carefully responded during the rebuttals.
The responses are clearly organized as responding to weaknesses and reviewer questions, and I don't think that any outstanding concern remains.

**Reviewer Scores:**

Reviewer uQyK increases their score during the discussions. If the review period was conducted as expected, I would expect the other reviewers increase their scores as well.

---

### Decision · Program_Chairs · 2026-01-26

Accept (Oral)